

# Direct or indirect recharge on groundwater in the middle-latitude desert of Otindag, China?

Bing-Qi Zhu[1*], Xiao-Zong Ren[2], Patrick Rioual[3]

[1]KLWCRESP, IGSNRR, CAS, Beijing, China

[2]SGS, TYNU, Jinzhong, China

[3]KLCGE, IGGCAS, Beijing, China

*Correspondence to*: Bing-Qi Zhu (zhubingqi@sina.com)

**Abstract.** The Otindag Desert in the middle-latitude desert zone of northern Hemisphere (NH) is essential to livestock-economy and ecoenvironment of northern China. Many areas in this zone are unexpectedly rich with groundwater resources although they have been under arid or hyper-arid climate for a long time. Widespread fresh groundwater deep to 60 m was found at the eastern part of the Otindag Desert. The occurrence of this massive fresh groundwater raises doubts on the long-lasting hypothesis in academic circles that regional atmospheric precipitation or palaeowater, namely the direct recharge, is the source of water in the middle-latitude desert aquifers of northern China. Understanding of the recharge of this fresh groundwater is important in evaluating the feasibility of groundwater exploitation and utilization. In this study we conducted hydrogeochemical and isotopical analyses to assess possible origin and recharge of these groundwaters. The analytical results indicate that the fresh groundwater is neither originated from regional atmospheric precipitation derived from the Asian Summer Monsoon system, nor from palaeowater that formed during the last glacial period. These findings suggest that the groundwater in this desert is possible to originate from remote mountain areas via the faults of the Solonker Suture zone, including the Daxing'Anlin and Yinshan Mountains. In addition, it is concluded that the hygeodrological linkage between desert aquifers and mountain systems through the suture zone is crucial to the hydrological functioning of the Otindag aquifer. This suggests that the modern indirect recharge mechanism, instead of the direct recharge and the palaeo-water recharge, is the most significant for groundwaterrecharge in the Otindag Desert. This study provides a new perspective into the origin and evolution of groundwater resources in the middle-latitude desert zone of HA.

**Keywords:** fresh groundwater recharge; atmospheric precipitation; direct recharge; indirect recharge; palaeowater recharge; fault hydrology; middle-latitude desert; Otindag Desert.

## 1. Introduction

The deficit of rainfall occurs globally in semi-arid to arid regions. It is usually made up by extracting groundwater to supply the needs of a growing population and a higher standard of living, Many areas in the middle-latitude desert zone of northern China such as the Badanjilin Desert, the Mu US sandy Land and the Hobq Desert (Chen et al., 2012a; Chen et al., 2012b), are unexpectedly rich



with large groundwater resources although they have been under arid or hyper-arid climate for a long
time (Sun et al., 2010). How these groundwaters originated and how they are recharged in these deserts
are thus fundamental scientific questions. Until now, however, no consensus has been achieved in
academic circles.

The Otindag Desert is one of the largest sandy lands located at the monsoon margin of northern

China and is the geographical centre of the northeastern Asian Continent (Fig. 1), which can be
regarded as a significant repository of information relating to the groundwater recharge in the arid
Inner Asia. At present, the eastern Otindag is also a typical case for its unexpected groundwater
resources, because there is abundant groundwater in this desert land and even rivers originate there due
to the spillover of spring water, such as the tributaries of Xilamulun River in its north and the Shandian
River in its south (Fig. 1). Climatically, the monsoon margin of northern China refers to a strip along
the present East Asian Summer Monsoon (EASM) limits and is considered to be sensitive to climate
change (Wang and Feng, 2013). Geologically, the Otindag Desert lies in a tectonic depression of the
central Solonker suture zone with a few faults stretching east and west (Fig. 2), with its northern
margin along a fault marked by a series of lake basins. Thus, the large-scale hydrogeological conditions
of the Otindag Desert belong to a fault zone under the influence of the EASM climate.

Until now, however, whether the climate or other factors affected the groundwater recharge in the

Otindag is still not known. Little data about the groundwater and its origin is available in the literature,
and knowledge and reliable data on various hydrogeological characteristics of the desert such as the
catchment extent, input/output, the hysteretic hydraulic functions, the transient hydraulic conditions,
in-homogeneities, and on transfer functions to overcome scale problems are also missing. Under such
conditions, conventional methods such as water balance and hydraulic methods sometimes fail in
determining groundwater recharge, particularly in extreme environments (arid, semi-arid, or cold)
(Drever, 1997). Because pristine aquatic conditions may significantly differ from managed conditions
in arid environment, and thus groundwater recharge is not a fixed number, but may vary with the
boundary conditions of the recharge system (Seiler and Gat, 2007).

Groundwater recharge can be broadly classified into two categories: the direct recharge by native

water resources and the indirect recharge by external water resources (Herczeg and Leaney, 2011).
Water infiltration of atmospheric precipitation through the unsaturated zone to the groundwater is
hydrologically defined as the direct recharge, and the indirect recharge is defined as recharge from
mappable features such as rivers, canals, lakes and originates from remote areas (Scanlon et al., 2006;
Healy, 2010). It is well known that groundwater recharge can be influenced by environmental factors,
including climate change, underlying soil and geology, land cover and the growth in human population
that affects withdrawal and economic development (Zhu et al., 2015, 2017). Among these
environmental factors, climate and land cover largely determine precipitation and evapotranspiration,
whereas the underlying soil and geology dictate whether a water surplus (precipitation minus
evapotranspiration) can be transmitted and stored in the subsurface (Doll, 2008, 2009; Giordano, 2009).

For some earth scientists, the direct recharge is thought to be very important for groundwaters in

the wide desert lands of north China due to the lack of surface runoffs (Yang et al., 2010; Yang and
Williams, 2003; Zhao et al., 2017). They argued that although the amount of atmospheric precipitation



is small, the vast catchment area in the desert region could concentrate the rainfall into large inland
basins, creating an aquifer with large storage capacity and great thickness. However, some hydrologists
estimated by the chloride mass balance method that the direct recharge was 1.4 mm/year, which
represents approximately only 1.7% of the mean annual precipitation in a cold large desert (Badanjilin)
in northern China (Gates et al., 2008). A similar estimation of 1 mm/year was given for Gobi deserts
from the Hexi Corridor to the Inner Mongolia Plateau in northwestern China (Ma et al., 2008).
Consequently, they thought that heavy potential evaporation and little precipitation make it difficult for
direct recharge to meet the supply of groundwater in these desert areas. Thus, the indirect recharge is
considered to be an important mechanism for groundwater recharge in these desert areas. For example,
Zhao et al. (2012) suggested that little precipitation had recharged into groundwaters in the Badain
Jaran Desert. Chen et al. (2004) argued that the groundwaters in the Badanjilin Desert were recharged
by palaeo-glacial melt water through faults and deep carbonate layers far away from the local desert.
Many studies also suggested that palaeowaters stored in an aquifer during wetter climate periods could
recharge to groundwater under certain conditions in arid lands (Edmunds et al., 2006; Ma and Edmunds,
2006). Other kinds of indirect recharge, such as mountain front recharge from adjacent mountain
blocks, are also proposed to offer an important inflow to aquifers within arid to semiarid catchments
(Blasch and Bryson, 2007).
In this paper, we focus to answer the question that whether groundwater recharge in Otindag is
mainly direct or indirect, using hydrochemical and isotopic indicators as tracers to offer a valuable
support for identifying the contributions of preicipitation recharge on groundwater, since these
indicators reflect the composition of water molecules and are sensitive to physical processes such as
mixing and evaporation (Sultan et al., 2000; Guendouz et al., 2003; Petrides et al., 2006; Scanlon et al.,
2006; Zhu et al., 2007, 2008; Jobbágy et al., 2011). The detailed objectives are: (1) to recognize the
major sources of groundwater in the area, and (2) to identify the key mechanism of groundwater
recharge in the desert.

**2. Regional settings**
Geographical location. The Otindag Desert lies between latitudes 42° and 44°N and longitudes
112° and 118° E (Fig. 1). It is an east part of the great middle-latitude desert zone between
northwestern and northeastern China which extends from the Taklamakan Desert in northwestern China
to the Kelqin Desert in northeastern China, near the west coast of the Pacific Ocean. The desert has an
area of approximately 21,400 square kilometers located in the eastern Inner Mongolia and at the
monsoon margin of northern China (Fig. 1). It is the fourth largest sandy lands in China (Yang et al.,
2012) and is bordered by a flat steppe terrain of Dali Basin to the north, the Yinshan Mountains and
mountainous loess landscape to the south, and the the Greater Khingan (Daxing'Anling) Mountains to
the east (Fig. 1). The Otindag Desert is essential to livestock-economy and ecoenvironment of northern
China. Settlements in this desert are constrained to oases to frequent springs, groundwater with high
level and areas where cultivation and irrigation are feasible. Some herdsmen live a precarious life by
grazing livestock in the desert.
Topography and geomorphology. The relief in the Otindag Desert is varied with a combination of



extensive dune fields and rugged piedmonts and mountains along the eastern and southern rims. In the
east, the Daxing'Anling Mountains has an average elevation ranging from 1,100 to 1,400 m and extend
from the Heilong River Valley into the upper reach valleys of the Xilumulun River from northeast to
southwest, with a gradual increase in height northwards from about 180 m near Huma to
Huanggangliang, where the highest mountaintop reach 2,029 m. In the south and southeast, the Yinshan
Mountains decline gradually near Duolun and Zhenglanqi, and in some areas leave wide alluvial plains.
The terrain of the Otindag Desert is less rough and elevations decrease from ca. 1300 m in the
southeast to ca. 1000 m in the northwest. Over the greater part of this desert the ground cover consists
of fixed and semi-fixed sandy dunes, with a few mobile dunes in area of little vegetation. The
dominated dune types are represented from parabolic to barchans, linear and grid-formed types,
ranging from a few meters to over 40 m in height (Zhu et al., 1980; Yang et al., 2008).
Climate, vegetation and soil. The climate of the Otindag Desert was not uniform in geological
period, with much sand movement, occasional rainy years, and several wetter intervals during the
Holocene (Yang et al., 2015; Tian et al., 2017). At present the whole desert belongs to the arid and
semi-arid temperate zone, with a mean annual temperature of 2 °C in the north and 4°Cin the south
(Liu and Yang, 2013). At the regional scale, the climate of the desert is typically controlled by the East
Asian Monsoon system, characterized by a warm summer, with precipitation transported by the EASM,
and by a cold and dry winter under the influence of the East Asian Winter Monsoon (EAWM). The
rainfall in the desert exhibits a wide variation in space and time. Influence of the EASM changes from
southeast to northwest in the desert, varying with the distance increase from the Pacific Ocean and
leading to the mean annual rainfall decreasing from ~450 mm in the southeast to ~150 mm in the
northwest (Yang et al., 2013). The spatial inequality of rainfall makes a great impact on the availability
of near-surface moisture, consequently on the distribution of vegetation, soil and the animal husbandry
potential of local communities. The major soil type is the grey desert soil in the west and changes to the
sierozems and chernozem or chestnut soil in the east. Through the desert, vegetation is sparse in the
west and relatively abundant in the east. The native vegetation is scrub woodland in the east and is
steppe in the west, showing a natural characteristic of the temperate desert or semi-desert. It is greatly
affected by temperature, rainfall and elevation in the growing season due to the scarcity of surface
runoff.
Geology. The Otindag Desert is located in a tectonic depression of the Solonker Suture Zone (Jian
et al., 2010) bounded by the Northern Early to Mid-Paleozoic Orogen Zone and the Hatug Uul Block to
the north, the Southern Early to Mid-Paleozoic Orogen Zone and the North China Craton system to the
south (Fig. 2). A few faults such as the Xar Moron Fault and Chifeng-Bayan Obo Fault stretch east and
west, with its northern margin along the Solonker Suture Zone marked by a series of lake basins (Figs.
1 and 2). The tectonostratigraphic units and overall structural trends are mainly oriented NE–SW (Fig.
2), which may be interpreted as resulting from overall compressive stresses oriented principally in the
NW–SE quadrants during orogenesis (Jian et al., 2010; Zhang et al., 2015). Diverse rock types from
unlithified and lithified clastic sediments through to carbonate, crystalline, and volcanic rocks are
distributed in and around the Otindag Desert (Zhang et al., 2015) (Figs. 2 and 3). Tertiary and
Quaternary sandstones and mudstones are the common basement rocks under the dunes of the Otindag,



and extensive volcanic basalts forming flat terrains are to the north (Zhu et al., 1980; Li et al., 1995).
Hydrology and hydrogeology. The Otindag Desert originated during the Late Quaternary (Yang
et al., 2015) and various alluvial fans formed at the margins of this desert during the early to middle
Holocene. These are composed of conglomerate and sand deposits, where major periodic steams or
wadis debouched into the Otindag. At present two rivers run through the eastern margin of the Otindag
Desert, i.e. the Xilamulun River in the north and the Shandian River and its two tributaries, the Shepi
River and Tuligen River in the south. Both stem from the eastern and southeastern parts of the Otindag
(Fig. 1). The Xilamulun River, 380 km in length and $32.54 \times 10^3 \, km^2$ in area, is a neighboring river both
to the northeastern Otindag and the southeastern Dali Basin, the northern catchment of the Otindag
Desert. The Xilamulun River flows to the east and finally goes into the Xiliao River, with an annual
mean runoff of $6.58 \times 10^8 \, m^3$  (Wu et al., 2014). The Shandian River is the upper reach of the Luan
River, with a length of 254 km and a catchment area of $4.11 \times 10^3 \, km^2$  (Yao et al., 2013). Spotted salt
crusts can extensively develop on land surface due to the high rate of evaporation. Sabkhas and salt
pans often form in areas surrounding the flat shorelines of some lakes in the Otindag. During rainy
season, some rain and floodwaters (generally coming from the Yinshan piedmonts) are retained in
low-lying areas, which may temporarily recharge shallow aquifers. Under storm conditions,
fast-flowing floods often form in some wadi channels with rich soil due to the occasional short, heavy
rainstorms.
Groundwater resources in the Otindag Desert and its surrounding areas depend on several kinds of
aquifers with different water-bearing formations and units (Fig. 3). Coarse- to fine-grained sedimentary
rocks, magmatic rocks and metamorphic rocks of the Inner Mongolia-Daxing'Anling Orogenic Belt
(Zhang et al., 2015) form the major regional aquifer unit (Fig. 3). They are composed mainly of alluvial
sediments (mid-Permian Zhesi Formation), melange (Solonker suture zone), A-type granite (early
Permian), bimodal volcanic rocks with sedimentary intercalations (early Permian Dashizhai Formation),
diorite-quartz diorite-granodiorite rocks (Carboniferous-Permian) and metamorphic complex
(predominantly gneiss, early Paleozoic) (Fig. 2). The aquifer is generally unconfined in dune fields of
the Otindag Desert, unconfined to semi-confined in the Yinshan Mountains' piedmont, and
semi-confined to confined in the Daxing'Anling uplands (Fig. 3). Water-level measurement in June
2010 indicated that the general depth of unconfined groundwater level ranges between 10 to 70 m in
the Otindag Desert (Fig. 3). Local granular aquifers in the central desert are composed of coarse fluvial,
lacustrine and aeolian sediments, but their extent and thickness vary throughout the watershed (Zhu et
al., 1980; Li et al., 1995). The generally coarse-grained texture of the unconsolidated rock formations
provides primary porosity in terms of groundwater flow in the desert.

**3. Methods**
The isotopes and ion chemistries of different water samples in the Otindag Desert, including
natural samples collected from local and regional precipitation, depression springs, shallow and deep
aquifers, perpetual lakes and outflowing rivers, are analyzed here and discussed. Relationships between
the study area and the regional prevailing EASM climate, the dominant topographical, geological
(tectonic) and hydrogeological conditions, are also explored and interpreted, using multiple graphs and





diagrams. Fieldworks took place during the summer season of 2011 and the spring season of 2012.
Water samples were mainly retrieved from shallow and deep wells located over a wide area in dune
fields of the study regions. The detailed locations of the sampling sites are shown in Fig. 4.
In this study, we designed two groups of parameters to characterize the physiochemistry of each
water sample. One is the field-measured parameters and another is the lab-measured parameters. The
former includes those parameters that will change in a shorter period of time when they are not directly
measured in the field, such as the total dissolved solid (TDS, mg/L), electrical conductivity (EC in
micro-Siemens per centimeter or µS/cm), hydrogen-ion concentration (pH) and temperature (°C). The
analysis for major cations (F$^-$, Cl$^-$, NO$_2^-$, NO$_3^-$, SO$_4^{2-}$HCO$_3$-, CO$_3^{2-}$and H$_2$PO$_4^-$) and anions (Li$^+$, Na$^+$,
NH$_4^+$, K$^+$, Mg$^{2+}$ and Ca$^{2+}$) are determined for all of the water samples collected. Contents of stable ($^2$H
and $^{18}$O) and radioactive isotopes ($^3$H) in the rain and groundwater samples are precisely measured. The
analytical data of the physiochemical parameters and the stable and radioactive isotopes of the water
samples collected in this study are listed in Tables 1, 2 and 3, respectively.

**4. Results and Discussions**

**4. 1. Hydrochemical characteristics of natural waters**
The natural water samples collected in this study are generally neutral to slightly alkaline, with the
pH values varying between 6.26 and 9.44 (except the precipitation sample p1, 4.61) (Table 1) and a
median value of 7.27. The TDS values range between 67 and 660 mg/L (average 211 mg/L) (Table 1),
all belonging to fresh water(TDS < 1000 mg/L) in the salination classification of natural water
(Meybeck, 2004). The variations in ion concentrations of the major cations and anions in the studied
water samples were displayed in a fingerprint diagram with a semi-logarithm y-axis (Fig. 5). The rain
water sample is the most depleted in ions among these samples. The groundwater samples have the
highest concentrations of cations and anions and the lake, river and spring waters had intermediate
values. The calcium concentration is the highest among cations in almost all of the water samples, and
the HCO$_3$+CO$_3$ concentration (bicarbonate + carbonate, alkalinity) is the highest among anions in most
of the water samples. For several groundwater samples (g3, g4, g5, g6 and g11), spring sample (s1) and
precipitation sample (p1), they have higher SO$_4$ concentrations than alkalinity (Fig. 5).
Two chemically distinct water types are recognized for the studied waters via a Piper diagram (Fig.
6), calcium bicarbonate and calcium sulphate. No Chloride-type and sodium-type waters occur in the
study area (Fig. 6). It has been reported that the global groundwater tends to evolve chemically towards
the composition of seawater (Chebotarev, 1955), and this evolution is associated with regional changes
in dominant anions but not cations. This general evolution of groundwater can be illustrated as a anion
evolution line (Freeze and Cherry, 1979): HCO$_3^-$ → HCO$_3^-$ + SO$_4^{2-}$ → SO$_4^{2-}$ + HCO$_3^-$ → SO$_4^{2-}$ +
Cl$^-$ → Cl$^-$ + SO$_4^{2-}$ → Cl$^-$, which travels along the flow paths and increasing ages. It can be deduced
from this line that bicarbonate water is the early product of groundwater evolution with low salinity ,
renewable water resources or low residence time, while sulfate waters is the intermediate or advanced
product of groundwater evolution with higher salinity passing through gypsum and anhydrite aquifers
(Clark, 2015). The distribution pattern of water chemical types occurred in the study area indicates a





primary stage of groundwater evolution in the Otindag Desert.

The δD values of the groundwater samples collected in this study varied from -63.42‰ to -75.92‰

(Table 3), with an average -69.53‰. The $\delta^{18}O$ values ranged between -8.64‰ and -11.26‰ (Table 3),
with an average -10.17‰. The spring water samples were relatively concentrated in δD and $\delta^{18}O$ and
were greatly similar to those of the groundwater samples (Fig. 7). The δD and $\delta^{18}O$ values in the river
water samples were slightly more variable and were also similar to those of the groundwater (Fig. 7).
The lake water samples were enriched in δD and $\delta^{18}O$ by comparison to the groundwater samples (Fig.
6). The precipitation sample p1 was also enriched in δD and $\delta^{18}O$ by comparison to the groundwater
samples (Fig. 7). The content of radioactive isotope of tritium ($^{3}H$) measured in seven well
groundwater samples with 6-60 m depth ranged from 1.86 to 24.35 TU (Table 3), with an average
14.95 TU, higher than the mean tritium concentration (9.8 TU) of groundwater in the Vienna Basin,
Austria (Stolp et al., 2010), the seat of the International Atomic Energy Agency (IAEA).

If we plot the relationships between oxygen and hydrogen isotopes of groundwater, spring, river

and lake water samples, we observed that most of the data points fell on a straight line that can be
expressed by a regression equation: $\delta D = 4.09\delta^{18}O - 28.31$ ($R^2=0.93$, n=24) (EL1 in Fig. 7). This local
groundwater line (LGWL) is different from the Global Meteoric Water Line (GMWL, $\delta D = 8\delta^{18}O +10$)
and the Mediterranean Meteoric Water Line (MMWL, $\delta D = 8\delta^{18}O +20$) estimated by Craig (1961), but
it is similar to the local groundwater lines established for other deserts in northern China and central
Asia with a same slope but different Y-intercepts, such as $\delta D = 4.17\delta^{18}O - 31.3$ for the Badanjilin
Desert (Jin et al., 2018), $\delta D = 4.8\delta^{18}O - 15.2$ for the Ejina Desert in China (Wang et al., 2013), and δD
$= 4.26\delta^{18}O + 9.23$ for the Rub Al Khal Desert in the United Arab Emirates (Rizk and El-Etr, 1997). The
data points are scattered for the lake water samples (Fig. 7) in the Otindag, suggesting that the lake
waters are affected by evaporation, but the other waters in the desert are not so.

**4. 2. Precipitation recharge on groundwater in the Otindag**

In order to compare the isotopic signals between groundwater and precipitation at a regional scale,

the isotopic analysis of precipitation from similar areas surrounding the study area, such as Baoutou,
were incorporated with local data of precipitation (p1) in this study (Fig. 7). The Baotou station is the
nearest long-term station to the Otindag Desert and was monitored for the isotopic composition of
rainfall for the period 1986-2001 within the International Atomic Energy Agency Global Network of
Isotopes in Precipitation (IAEA-GNIP) database. The stable isotope data from Baotou was used to
represent the regional background of stable isotopic compositions of the present-day meteoric water,
especially in the westward inland areas of the Otindag Desert (Fig. 1). In addition, stable isotope data
of the Tianjin station was also used to represent the regional background of precipitation in the eastern
coastal areas of the Otindag Desert (Fig. 1).

Based on the isotopic data from the Baotou station, the local meteoric water lines can be

statistically expressed as the isotopic regression equation of $\delta D = 6.36\delta^{18}O - 5.21$ (LMWL-B). It can
also be expressed as $\delta D = 6.57\delta^{18}O + 0.31$ (LWML-T), based on the data from the Tianjin station (Fig.
7). The precipitation sample p1 collected in this study fell onto the GMWL (Fig. 7). It also showed
similar δD and $\delta^{18}O$ values to those of the precipitation collected in the GNIP stations of Baotou and



Tianjin (Fig. 7).

Compared to the precipitation data from the GNIP stations and from the local precipitation (p1),

the groundwater, spring, and river water samples were evidently depleted in heavy stable isotopes in
the Otindag (Fig. 7). Except for the lake water samples, most of the groundwater, river water and
spring water samples in the Otindag fall on or lay between the LMWL-B and the LMWL-T lines, and
are located at the lower left area of the precipitation points (Fig. 7).

Because the isotopic evolution of δD and δ$^{18}$O in water illustrated in the Craig line represents a

one-way and irreversible process, the water bodies distributed at the upper right area of the Craig line
can not be recharge sources for the water bodies distributed at the lower left area of the line. Such
results indicate that the groundwater, river water and spring water in the Otindag are not recharged by
the regional precipitation, namely no significant modern direct recharge has taken place for
groundwater in the Otindag.

Dogramaci et al. (2012) documented that only intense and remarkable rainfall events >20 mm

could recharge groundwater in the semi-arid Hamersley Basin of northwest Australia, while the rainfall
events <20 mm had limited influences on groundwater recharge. Chen et al. (2014) described that
rainfall events ≤5 mm in the arid and semi-arid region of northern China would be evaporated into
the atmosphere rapidly before it is infiltrated into the groundwater system. Based on the analysis on the
data records from two meteorological stations around the Otindag, i.e.the Duolun and Xilinhaote
stations (see Fig. 1a), we observed that rainfall events >20 mm on average only occur 2.5-3.4 times per
year (Table 4). In some years (e.g. from 2005 to 2007 at the Xilinhaote Station), no rainfall events >20
mm even occurred. It further indicated the limited contribution of regional precipitation on
groundwater recharge in the Otindag.

In addition to groundwater, the river and spring water samples from the Otindag also deviated

from the local precipitation in the Craig diagram (Fig. 7).These water samples came from the
Xilamulun, Shepi and Tuligen rivers. They shared the same evaporation line (EL1) with the
groundwater and lake water samples (Fig. 7). Generally speaking, natural waters that have a same
recharge source are distributed on a same line of evaporation in the δ$^2$ and δ$^{18}$O diagram (Chen et al.,
2012b). This indicates that the recharge sources of groundwater, river water, spring water and lake
water in the Otindag are genetically associated each other and differ from the local precipitation.

**4. 3. Winter precipitation and palaeowater recharge on groundwater in the Otindag**

Since the groundwater samples in the Otindag are depleted in their δD and δ$^{18}$O values even more

than those of the local rainfall (Fig. 7), they must be sourced from other waters characterized by similar
or more depleted signals in their stable isotopes compositions. Due to the temperature effect (such as
evaporation) on isotopic fractionation, only the waters issued from colder environments can be more
depleted in their δD and δ$^{18}$O values even more than those of the local rainfall.

Because the Otindag Desert is under the control of the EASM climate (Fig. 1), the local rainfall in

the desert is maily sourced from summer precipitation. This can also be illustrated by the seasonal
distributions in annual mean precipitation (Fig.8a), in annual mean air temperature (Fig. 8b) and in
annual mean water vapor pressure (Fig. 8c) over the last forty years at the two surrounding GNIP



weather stations in Baotou and Tianjin. The seanonal distributions of stable isotopes in the two stations (Fig. 8d-e) show that the summer rainfall is evidently positive in its signals of $\delta D$ and $\delta^{18}O$ by comparison with those of the winter rainfall, further suggesting that the waters issued from cold environments can be more depleted in their $\delta D$ and $\delta^{18}O$ values than those of the summer rainfall. Thus we speculate that groundwater in the Otindag can be potentially derived from (1) modern precipitation in winter, (2) palaeowater formed in the past glacial period, or (3) remote/mountains waters that emanate in colder and wetter conditions.

The annual mean values of $\delta D$ and $\delta^{18}O$ over the last forty years are more depleted in winter precipitation than in summer precipitation at the Baotou and Tianjin stations (Fig. 8d-e). This isotopic signal qualifies the regional winter precipitation to be a potential souce of groundwaters in the Otindag. However, the precipitation amounts and the water vapor pressures (effective moisture) in winter months are much lower than those in the summer months at both the Baotou and Tianjin stations (Fig. 8a and 8c). It indicates that the winter seasons in these regions are relatively colder and drier but not colder and wetter. A colder-wetter winter season is a necessary condition for winter precipitation to be a water source for the formation of groundwater under a summer monsoon climate. This is because the bigger amounts of summer precipitation will easily remove or weaken the depleted isotopic signals of winter prepicitation in groundwater. In this regard, modern winter precipitation is unlikely to be an important source of groundwater in the Otindag.

As to the palaeowaters formed in colder and wetter periods such as the last glacial, it has been proposed to be a potential water source for groundwaters in the wide arid lands of the world. The depleted signals of stable isotopes ($\delta D$ and $\delta^{18}O$) in groundwater have been recognized in global arid and semi-arid regions, such as the Sinai Desert in Egypt (Gat and Issar, 1974), Israel (Gat, 1983), South Australia (Love et al., 1994, 2000), northern China (Ma et al., 2010), Saudi Arabia (Bazuhair and Wood, 1996) and North Africa (Guendouz et al., 2003). These signals are very often explained as palaeo-groundwater that recharged by precipitation during past wetter and colder periods (Love et al., 1994, 2000; Herczeg and Leaney, 2011).

Here we use the tritium data as a environmental tracer to estimate the groundwater age in the Otindag. The tritium data at the GNIP stations of the Baotou and Tianjin are also referenced as the background values in precipitation of recent years. The residence time of groundwater in aquifer and the residual tritium of a water body can be calculated by $N = N_0 e^{-\lambda t}$ (Yang and Williams, 2003). Where N = content of residual tritium in water sample, $\lambda = 0.0565$, the radioactive decay constant, $N_0$ = content of tritium at the time of rainfall and t = years after precipitation. Based on this equation, the residual tritium was theoretically calculated and the standard for tritium dating was established for seven groundwater samples in the Otindag Desert (Table 3). As a result, ages of 0-60 years were obtained for these groundwater samples (Table 5). This indicates that recent recharge took place several decades after the peak in global nuclear tests. We thus conclude that groundwater is generally not older than 70 years in the study area. It means that groundwater in the Otindag are not palaeowater recharged.

Both the modern summer and winter precipitation recharge and the palaeowater recharge can be refuted, indicating that direct recharge is not a major mechanism controlling the groundwater recharge



in the Otindag.

### 4. 4. Remote water recharge on groundwater in the Otindag: Dali Basin

The third hypothesis that "remote/mountains waters emanate under colder and wetter conditions"
is further considered here. In essence, it is an indirect recharge mechanismas as water originates from
remote areas (Healy, 2010; Herczeg and Leaney, 2011).
It is worth noting that the values of deuterium and oxygen-18 for groundwater in the north part of
the study area are more depleted in $\delta D$ and $\delta^{18}O$ than those in the south part (Table 3). It suggests that
the Otindag groundwater might be potentially recharged by water resouces coming from the northern
neighboring catchment, such as the Dali Basin.
Recently published data of $\delta D$ and $\delta^{18}O$ in groundwater, lake water, river water and spring water
sampled from the Dali Basin (e.g., Chen et al., 2008; Zhen et al., 2014) were compiled in this study and
were co-analyzed with the data from the Otindag. About 70 natural water samples from the Dali and
Otindag with $\delta D$ and $\delta^{18}O$ values are shown in a Craig diagram (Fig. 9). All of these samples fell on or
lied near the evaporation line EL2 in the Craig diragram (Fig. 9), with a regression equation of $\delta D =$
$4.81\delta^{18}O$ - 21.55 and a high correlation coefficient ($R^2$=0.98, n=70). Compared to the groundwater
samples in the Otindag, water samples from the groundwaters, rivers and springs from the Dali Basin
are more depleted in $\delta^{18}O$ and $\delta D$ (Fig. 9). Such results further indicate that, in terms of itsisotopic
signature, the groundwater in the Otindag has a close relationship with the natural waters in the Dali
Basin.
The similar signals of $\delta D$ and $\delta^{18}O$ between the groundwater in the Otindag and the river water in
the Dali (Fig. 9) point towards the idea that the groundwater in the Otindag might be sourced from the
river water in the Dali Basin, since the Dali has more depleted isotopic signals in water than the
Otindag (Fig. 9). Considering the topographical gradient of elevations between the two regions,
however, river water in the Dali Basin cannot flow into the eastern Otindag, because the terrain
elevation of the Dali Basin is lower than that of the Otindag (Fig. 1). This is also the reason why the
huge Dali Lake that lies in the Dali Basin has no equivalent in the Otindag (Fig. 1). If there is a
hydraulic linkage between the two regions, water should flow from the Otindag into the the Dali, but
not conversely.
In view of the hydraulic gradient, river water in the Dali Basin could not be a recharge source for
groundwater in the Otindag. However, in view of the isotopic gradients, groundwater in the Otindag
could not conversely be the source of river water in the Dali (Fig. 9). Thus, the similar isotopic signals
between the river water in Dali and the groundwater in Otindag indicate that these waters might be
recharged from a common source.
Similar isotopic signals also occurred in the groundwaters between the Otindag and the Dali Basin
(Fig. 9). In order to understand the linkage of groundwaters between the two regions, the potential
movement of groundwater in the transition zone of the two regions need to be known. In this study, a
groundwater-sampling project was designed in the field along a N-S section of a palaeo-channel
located at the transition zone between the Dali and Otindag (Figs. 1, 2). The channel was named
"PCSX" in this study, with its north part named "NPCSX" and the south part named "SPCSX".



The GPS elevation of the northernmost sampling site in the NPCSX (g11, about 1317 m a.s.l.) was
much lower than that of the southernmost site in the SPCSX (g1, 1396 ma.s.l.) (Fig. 2 and Table 1).
Regarding to the topographical gradient in the channel, there is a drop of about 80 m between the
NPCSX and the SPCSX. Under such slope, the underground hydraulic gradient for groundwater flow
can be roughly parallel with that of the surface water flow, namely that the groundwaterflow should
move downwards from the SPCSX area into the NPCSX area. Thus we can speculate that groundwater
in the NPCSX would have higher salinity than those in the SPCSX under such flowing direction. In
order to verify this speculation, actual variations of water salinity (chloride and TDS) were detected
along the PCSX section. The sampling site g1 was defined as the initial point and the distances between
g1 and other sampling sites along the PCSX section were calculated, based on their GPS geographical
coordinates measured in the field. The results are shown in Fig. 10a-b. It is clear that the variations of
chloride and TDS concentrations in groundwater do not increase along the palaeo-channel from south
to north (Fig. 10a-b). On the contrary, both the values of chloride and TDS are lower in the NPCSX
area than those in the SPCSX area. Such kind of spatial variations in the chloride and TDS values
contradict the speculated patterns abovementioned, suggesting that the hydraulic gradient of
groundwater flowing path in this region is not controlled by the topographical gradient between the
NPCSX and SPCSX areas.
Compared between the NPCSX and SPCSX regions, the stable isotopic values ($\delta^{18}$O and $\delta$D) of
groundwaters in the SPCSX region vary greatly with a large amplitude, while those in the NPCSX are
relatively constant (Fig. 10c-d). The constant variations indicate that the recharge source of
groundwater in the NPCSX is relatively unitary. The isotopic values in the SPCSX are much lighter
than those in the NPCSX along the distance section from south to north (Fig. 10c-d). The heaviest
values occurred in the sample g11 collected from the NPCSX (Fig. 10c-d), indicating a water being
earlier recharged. The spring water sample s2, a representation of discharge water, is characterized by
medium values of $\delta$D and $\delta^{18}$O. These results indicate that the groundwaters in the SPCSX area, with
relatively enriched isotopic signals in $\delta$D and $\delta^{18}$O by comparison with those in the NPCSX area, are
composed of a mixture of the groundwaters in the NPCSX with other waters.
The tritium contents were broadly and positively related to the values of deuterium excess in the
groundwater samples in the PCSX (Fig. 10e). For water that experiences an evaporation process, the
d-excess value will increase in the evaporated water vapor, but will decrease in the residual water body
(Dansgaard, 1964; Merlivat and Jouzel, 1979). In this study, except for sample g11 (a sample very
close to the riverhead area), the positive relationship between the tritium and the deuterium excess
generally shows that the d-excess values are higher in the groundwaters collected from the NPCSX, but
are lower in those from the SPCSX (Fig. 10e). This distribution pattern indicates that the groundwaters
in the NPCSX are relatively younger and experienced a lower degree of evaporation than those in the
SPCSX. The d-excess gradient, increasing from south to north in the PCSX, further suggests that
groundwater does not flow from the SPCSX area to the NPCSX area, namely out of the topographical
control.
Many studies (e.g., Boronina et al., 2005; Kazemi et al., 2006) have demonstrated that
groundwater flows in the direction in which it gets older. In view of this point, groundwaters in the



PCSX region should flow from the NPCSX area to the SPCSX area, in opposition to the S-N
topographical gradient between the Otindag and Dali regions. Thus groundwater in the Dali are not the
source of groundwater in the Otindag. The similar isotopic signals between groundwaters in the two
regions indicate that these waters might be recharged from a common source in other place.

**4. 5. Remote water recharge on groundwater in the Otindag: mountains waters**

The discussions above revealed that both the groundwaters in the Otindag and DaliBasin might be

recharged from a common source derived from another place. Considering the third hypothesis
abovementioned that "remote/mountains waters emanate under colder and wetter conditions", we
propose that this "common source" of the two regions are from mountians areas surrounding the
Otindag and Dali Basin.

There are two large permanent rivers and lots of small intermittent streams entering the Dali Basin

(Xiao et al., 2008), including the Xilamulun River to the south and the Gongger River to the north, both
of which are stemming from the Greater Khingan Mountains (Daxing'Anling Mountains in Chinese
pinyin, 1,100-1,400 m above seal level) (Fig. 1). The Xilamulun River carries a large amount of water
(about $6.58 \times 10^8$ m$^3$/y) from the Daxing'Anling Mountains flowing through the east margins of the Dali
and Otindag (Wu et al., 2014). This is an important clue linking natural waters between the Otindag
and Dali Basin.

Variation in the elevation from the Dali Lake to the riverhead of the Xilamulun River can be

clearly found along a land surface topographical section (Fig. 11). The channel of the Xilamulun River
is located in the Xar Moron Fault (Fig. 1), which is a part of the Solonker Suture Zone (Eizenhöfer et
al., 2014) or the Xilamulun-Changchun-Yanji plate suture zone (Sun et al., 2004) in the regional
tectonical settings (Fig. 2). Outcrop observations indicate that fault zones commonly have a
permeability structure suggesting they should act as complex conduit–barrier systems in which
along-fault flow is encouraged and across-fault flow is impeded (Bense et al., 2013). Thus the
hydraulic grediant of groundwater flow in the Eastern margins of the Otindag and Dali Basin must be
controlled by the fault zone hydrogeology. This may be the reason why the hydraulic gradient of
groundwater represented by the isotopic and hydrogeochemical gradients of groundwater samples in
this study is not consistent with the local topographical gradient in the Otindag Desert. On the other
hand, the regional aquifer is generally unconfined in dune fields of the Otindag Desert but
semi-confined to confined in the Daxing'Anling uplands (Fig. 3), thus the thick unconsolidated
aquifers in the study area (Figs. 3 and 11) will be favourable conditions for groundwater storage and
transportation along the Solonker Suture Zone. When rivers stem from the Daxing'Anling
Mountainsand flow downward to the marginal areas of the Dali and Otindag, leakage water from these
rivers can recharge the desert land through thick unconsolidated aquifers. A strong isotopic evidence is
that the lake and river waters in the Dali Basin share the same evaporation line (EL2) with the
groundwaters in the PCSX area.

Although groundwaters in the SPCSX area are different from those in the NPCSX area, their

isotopic data points still fell onto the EL2 (Fig. 9), which further indicates that the groundwaters in the
SPCSX are a mixture of waters from the Daxing'Anling Mountain and other sources. Another source



for groundwater recharge in the SPCSX could be represented by remote water such as flash floods
coming from the north Yinshan Mountains, because it can be clearly observed from digitial maps that
many transient rivers or streams originated from the Yinshan Mountrains flow into the south and
southeastern Otindag (Fig. 1). Supportive evidence for this idea can also be observed in the summer
rainy season. During rainy days or under storm conditions, fast-flowing floods caused by occasional
short, heavy rainstorms can form in playas, wadi channels and low-lying depressions in the unconfined
to semi-confined areas of the Yinshan Mountains' piedmont. These waters may temporarily recharge
shallow aquifers in the SPCSX area.

**5. Conclusions**
In the middle-latitude desert zone of northern China, many deserts such as the Otindag and
Badanjilin Deserts, are unexpectedly rich in groundwater resources, although they have no surface
runoff and have been under an arid or hyper-arid climate for a long period of time. How groundwaters
originated and recharged in these deserts are thus key questions that are still under debate. For some
earth scientists, the direct recharge is thought to be very important for groundwaters in the wide desert
lands of northern China, due to the lack of surface runoffs. However, groundwater availability is very
much a function of the local- and regional-scale geological and climatic settings. To achieve an
integrated understanding of the groundwater recharge and its controlling mechanisms is of great
significance. In this study, groundwater recharge was explored using multiple environmental tracers in
the Otindag Desert of northern China, a region that is under the influence of the East Asian Summer
Monsoon (EASM) climate. Compared to modern summer precipitation, the groundwaters, river waters
and spring waters are depleted in $\delta D$ and $\delta^{18}O$. All these waters shared a same Craig line, indicating a
genetic relationship on their recharge sources. The stable isotopic signals of the groundwaters is more
depleted than those of the modern summer precipitation and this suggests that the groundwaters studied
could only be sourced from cold water different from the EASM precipitation. In general, the analyses
revealed that the highland remote water resources from the Daxing'Anling and Yinshan Mountains
were isotopically and geochemically traced to be a major source for the groundwater in the Otindag. It
suggests that the modern indirect recharge mechanism, instead of the direct recharge and the
palaeo-water recharge, is the most significant for groundwater recharge in the eastern Otindag. This
study provides a new perspective into the origin and evolution of groundwater resources in the
middle-latitude desert zone of northern China.

**Acknowledgements**
This study was financially supported by the National Natural Science Foundation of China
(41771014 and 41602196) and the National Key Research and Development Program of China
(2016YFA0601900). We thank the China Meteorological Data Sharing Service system for providing
the weather data. Sincere thanks are also extended to Profs. Xiaoping Yang, Xunming Wang, Jule Xiao
and other workmates, e.g., Ziting Liu, Hongwei Li, and DeguoZhangfor their generous help in the
research work.



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

Chinese).





**Figure Captions:**

**Fig. 1.** The Geographical location of the Otindag Desert in northern China. (a) The study area shown at a large scale, and (b) the study area shown at a smaller scale, with detailed information about the boundary and tectonic settings of the desert land. 1, the palaeo lake area of the megalake Dali; 2, the boundary of the Otindag; 3, the modern lake area; 4, the boundary of Fig. 2; 5, the boundary between the westerlies and the East Asian Summer Monsoon (EASM) climate systems. ①, the Xilamulun River. ②, the Gonggeer River. ③, the Shepi River. ④, the Tuligen River. The boundary between the westerlies and the EASM in (a) and (b) is modified from Chen et al. (2010). The palaeo lake area of the megalake Dali and the palaeo channel in (b) is modified from Yang et al. (2015). The location of the Xar Moron Fault is referenced from Eizenhöfer et al. (2014). Section S1 is an elevation section starting from the upstream of the Dali Lake and ending with a spring sample (s2) in the riverhead of Xilamulun River.

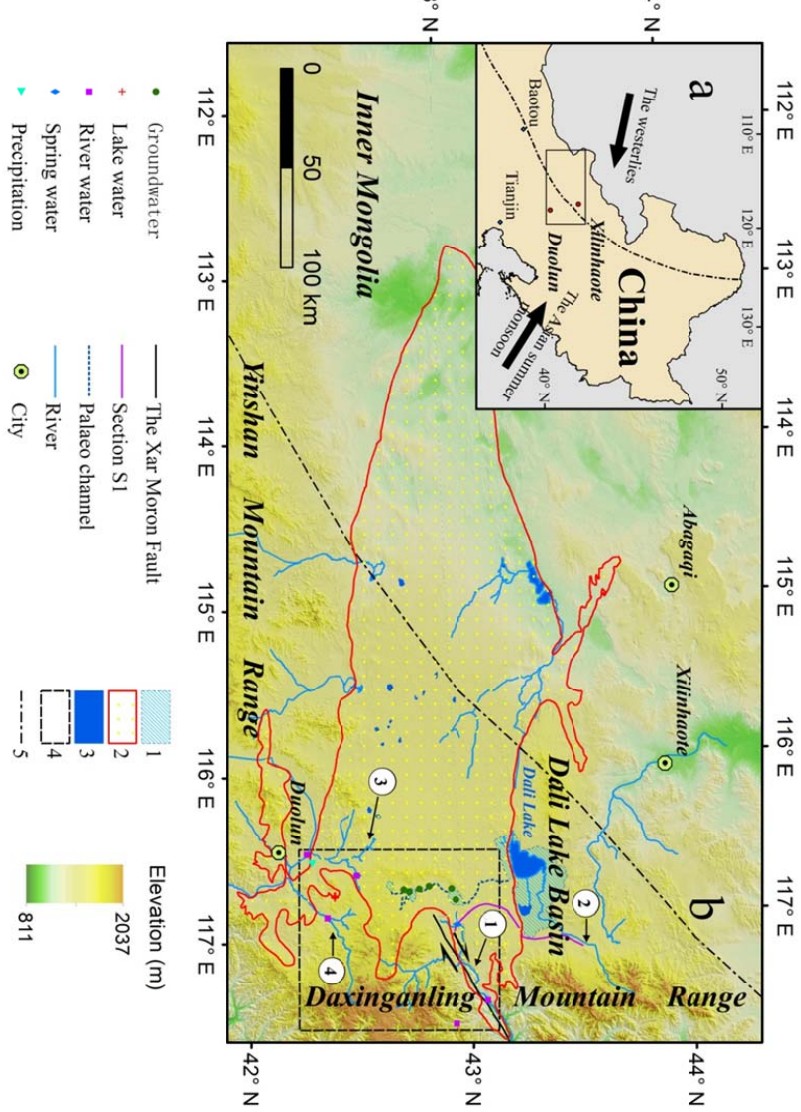

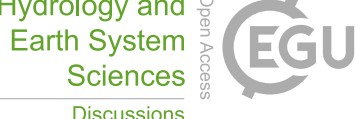



**Fig. 2.** (a) Tectonic framework of the north China-Mongolian segment of the Central Asian Orogenic
Belt (modified after Jahn, 2004). (b) Geological sketch map of the northern China-Mongolia tract
(modified after Jian et al., 2010). The Solonker suture zone represents the tectonic boundary between
the northern (Hutag Uul Block-Northern orogen) and the southern (southern orogen-Northern margin
of North China craton) continental blocks. Note that the red line marks the early Permian
paleobiogeographical boundary (Wang and Liu, 1986; Li, 2006), which coincides with the northern
boundary of the suture zone.

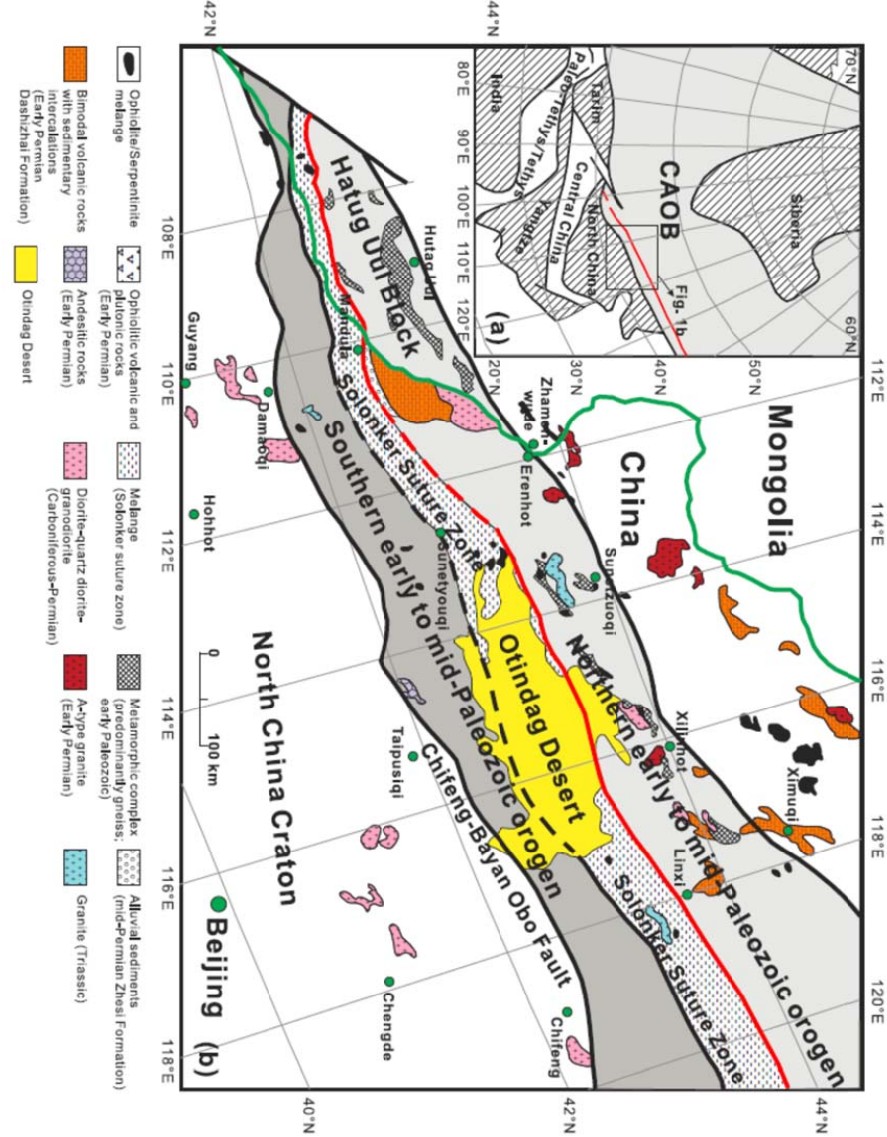





**Fig. 3.** The hydrogeological division map of the Otindag Desert.

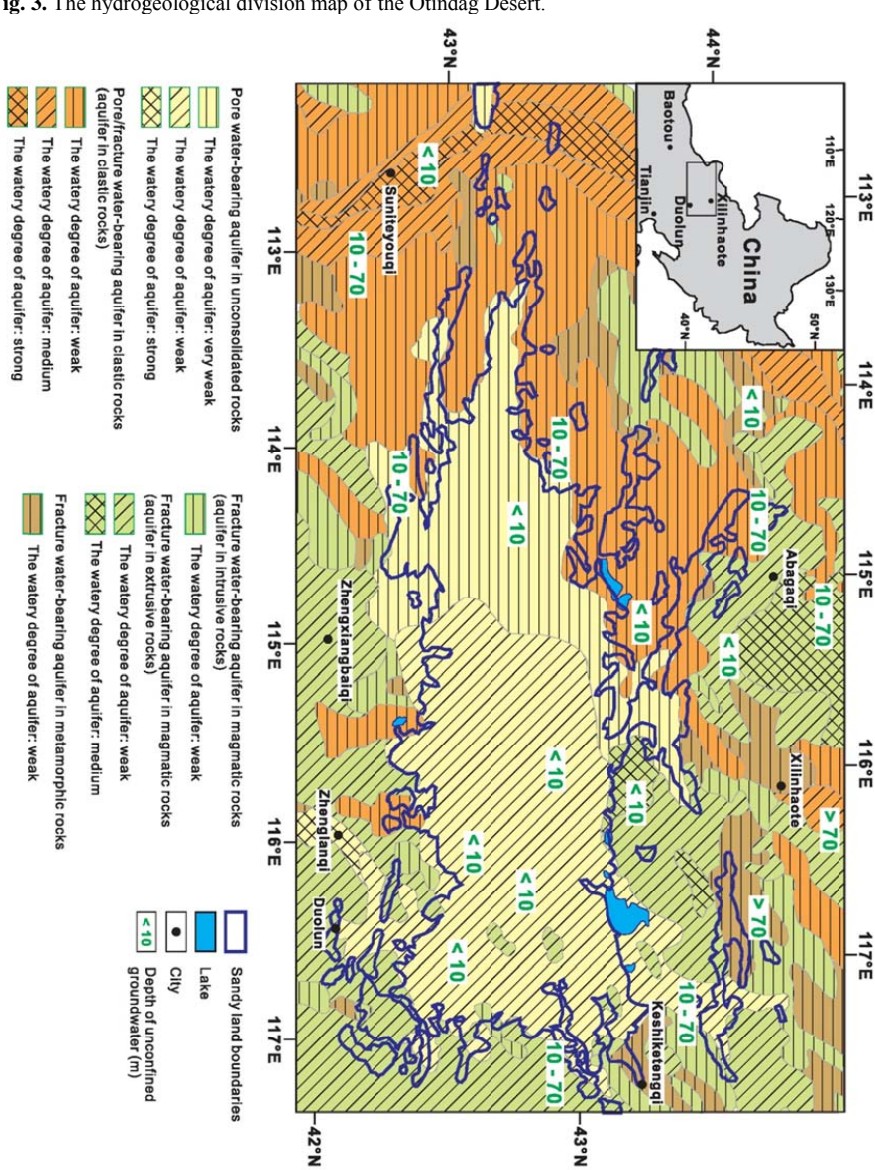






**Fig. 4.** The locations of the water sampling sites in this study.

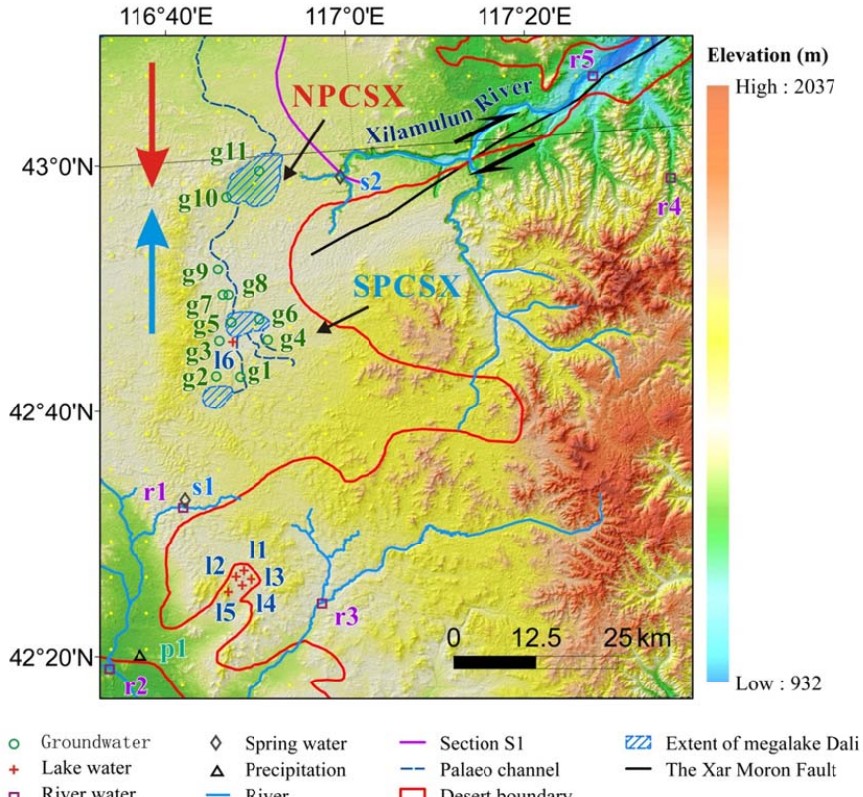






**Fig. 5.** The fingerprint diagram showing the variations of multiple ions' concentrations in the studied
water samples in an equivalent unit. The HCO$_3$+CO$_3$ concentration in the sample p1 was not shown,
due to its value being lower than the detection limit.

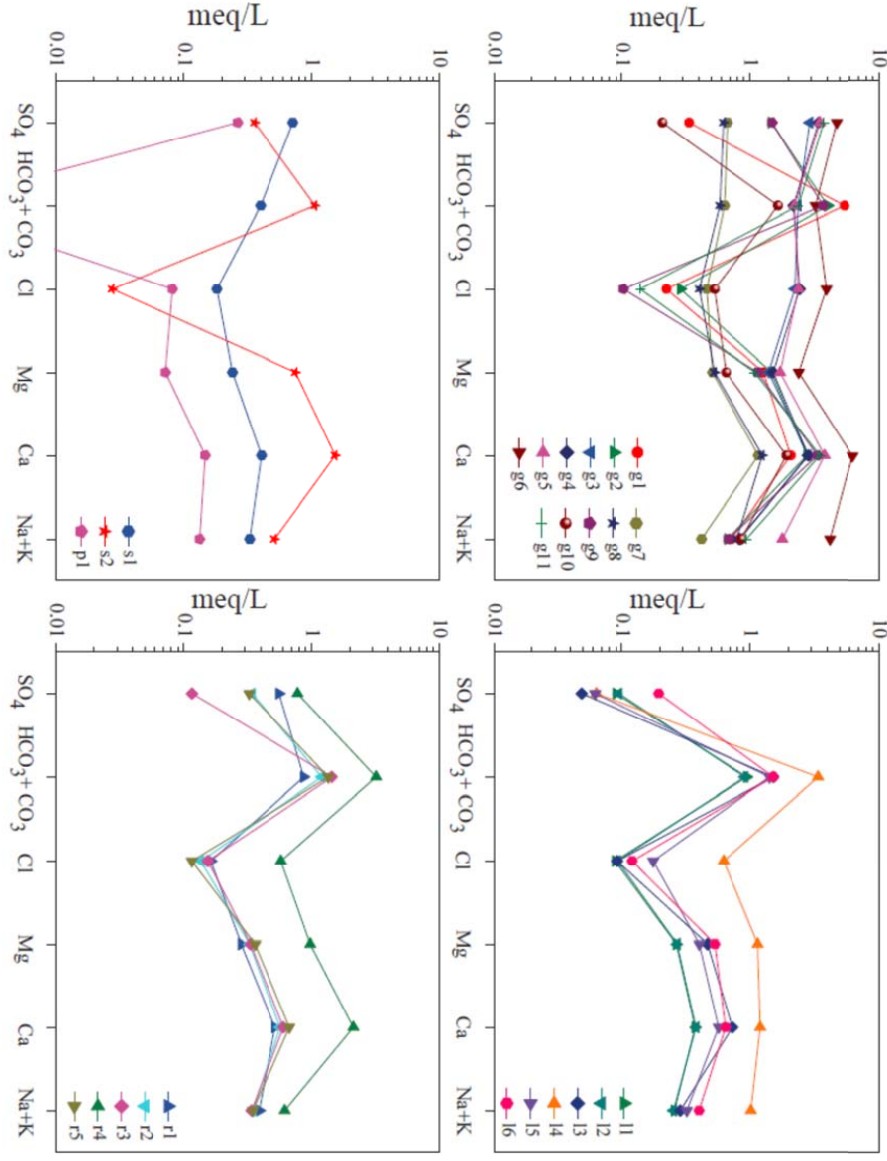






**Fig. 6.** The Piper diagram showing the relative abundances of major cations and anions in the studied
water samples. Major water types are also shown in this diagram.

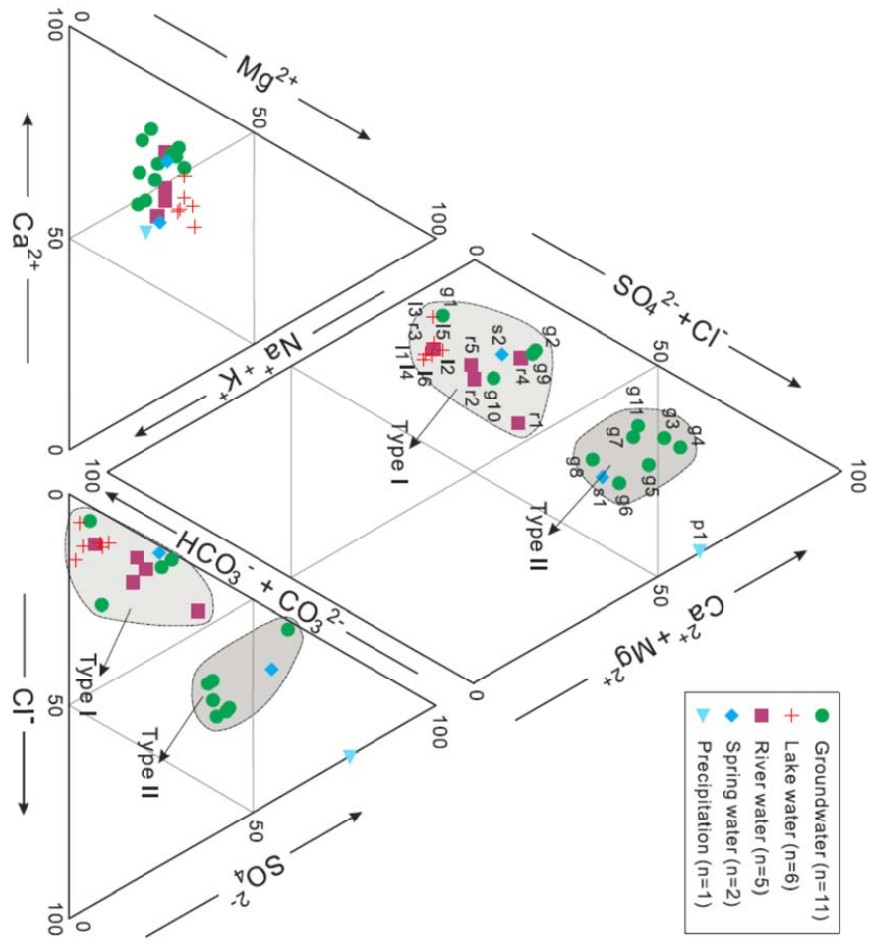






**Fig. 7.** The bivariate diagram of δD and δ¹⁸O, i.e. the Craig diagram, for the natural water samples in
this study. Different relationships between the groundwaters, lake waters, river waters, spring waters
and the precipitation waters are illustrated. AWMB, the annual weighted mean value at the Baotou
station; AWMT, the annual weighted mean value at the Tianjin station; LWMB, the long-term weighted
means at the Baotou station; LWMT, the long-term weighted means at the Tianjin station; GMWL, the
Global Meteoric Water Line; LMWL-B, the local meteoric water line calculated based on the data from
the Baotou station; LWML-T, the local meteoric water line calculated based on the data from the
Tianjin station; EL1, the evaporation line calculated based on the data of water samples collected in this
study.

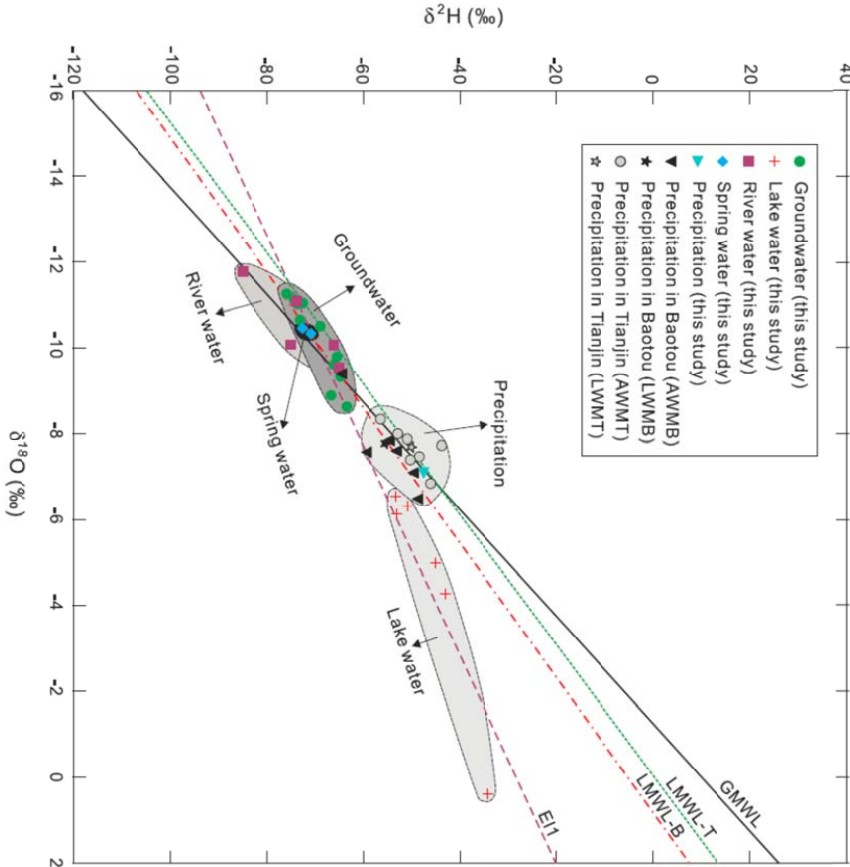






**Fig. 8.** The seasonal mean distributions of (a) precipitation, (b) surface air temperature and (c) water
vapor pressure from the Baotou and Tianjin weather stations (station sites seen in **Fig. 1a**) in the
surrounding areas of the Otindag for the period 1981-2010. The seasonal mean distributions of (d) $\delta^{18}$O
and (e) $\delta$D values in precipitation from the Baotou and Tianjin weather stations in the surrounding
areas of the Otindag for the period 1986-2001.

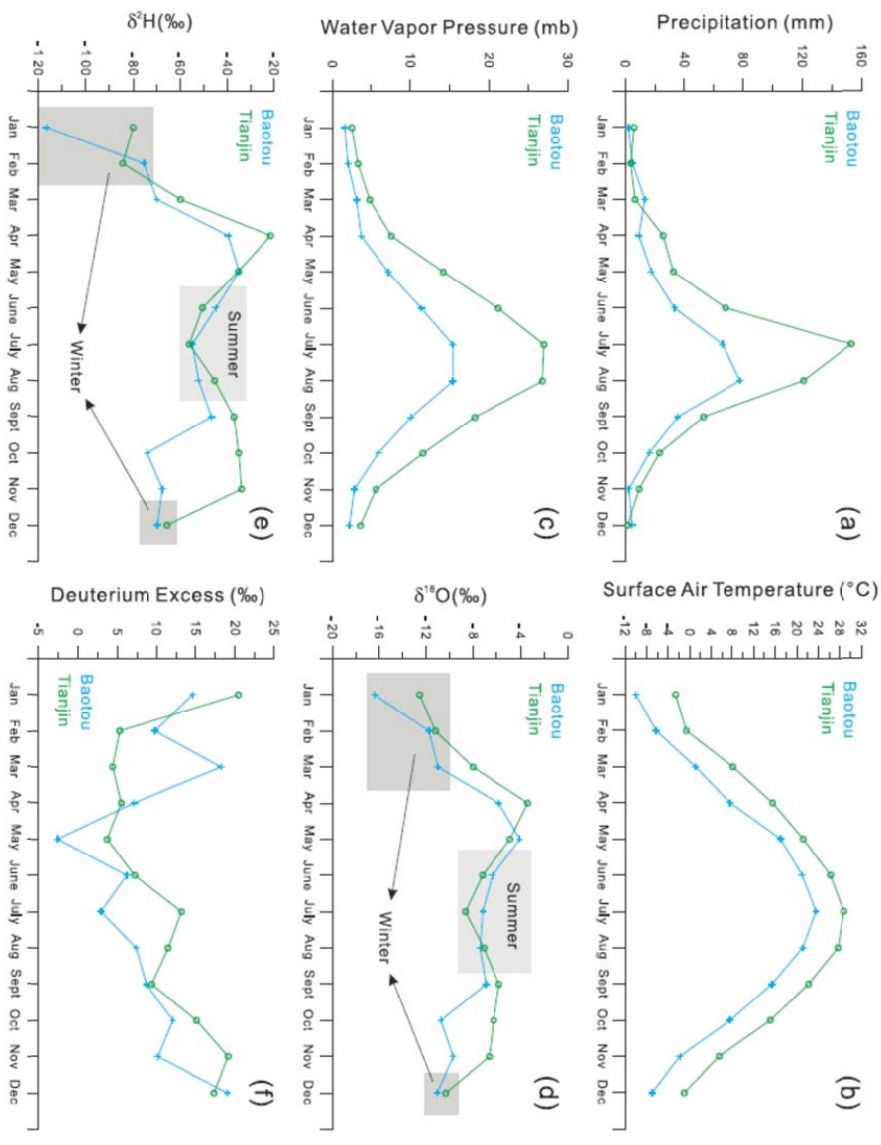





**Fig. 9.** The bivariate diagram of δD and δ¹⁸O, i.e. the Craig diagram, for the natural water samples
collected in the Otindag (this study) and the Dali Basin. Different relationships between the
groundwaters, lake waters, river waters, spring waters and the precipitation waters are clearly
illustrated. AWMB, AWMT, LWMB, LWMT, GMWL, LMWL-B, LWML-T, and EL1 are the same as
in Fig. 7. EL2, the evaporation line calculated based on the data from the groundwater, lake water, river
water and spring water samples collected from the Otindag and Dali Basin. The data for the Dali were
taken from Chen et al. (2008) and Zhen et al. (2014).

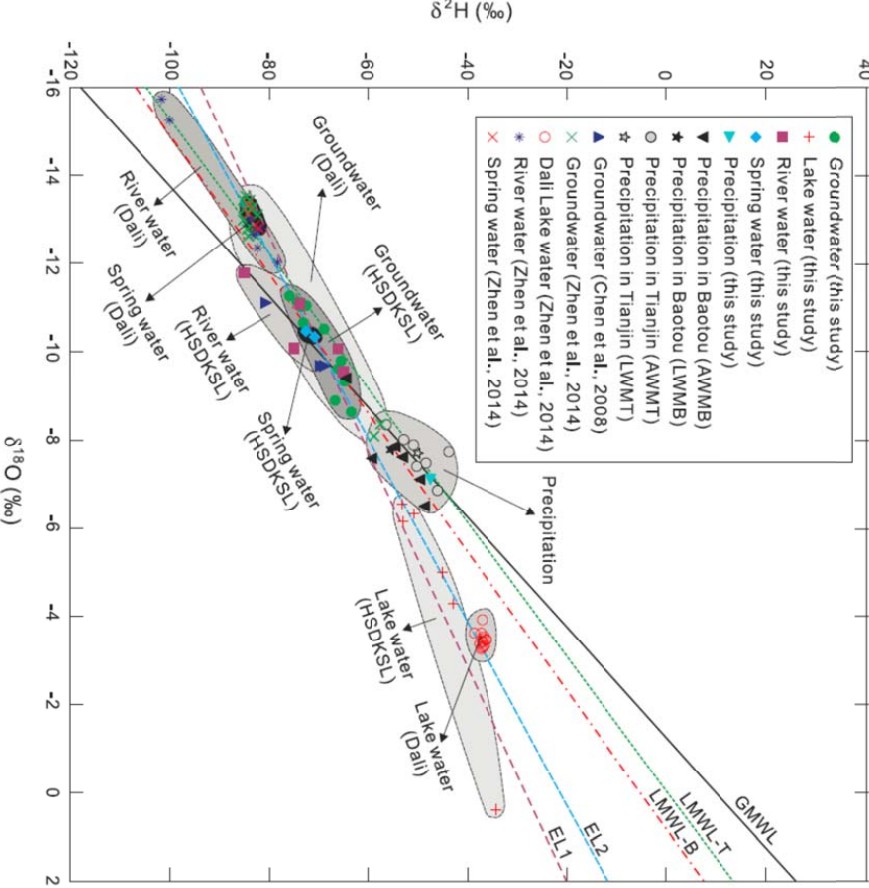






**Fig. 10.** (a) Sketch map showing the relationship between the groundwaters in the NPCSX and SPCSX
areas, based on variations of (a) the chloride concentrations, (b) the TDS concentrations, (c) the $\delta^{18}$O
values and (d) the $\delta$D values of these water samples versus their distances away from the water sample
g1 along the palaeo river channel (PCSX) from south to north. The dashed line in (c) and (d) represents
the corresponding values of the spring water sample s2, and divides samples into the NPCSX and
SPCSX parts. (e) Variations of tritium contents vs. deuterium excess for the groundwater samples in the
study area. The sample g6 was omitted due to its potential contamination.

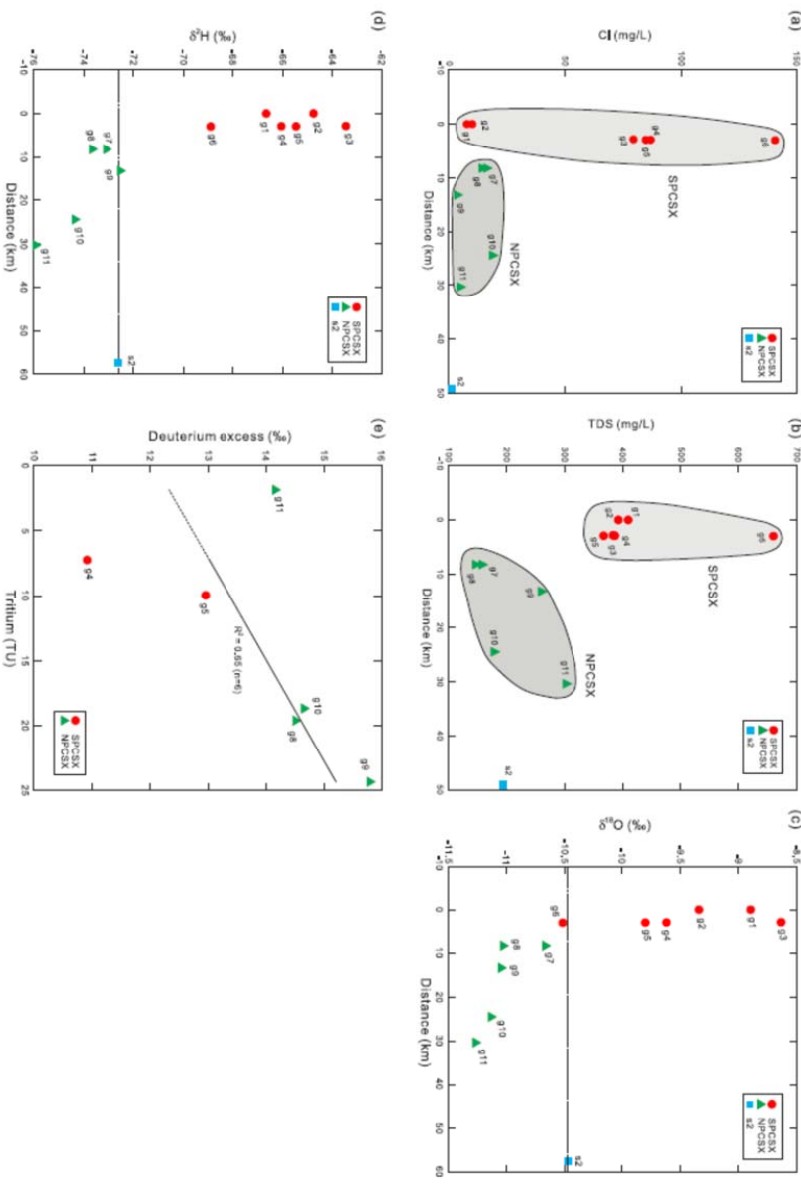




**Fig. 11.** Variation of the topographical elevation along the section S1 (see Fig. 1b) from the upstream of
the Dali Lake to the location site of the spring water sample (s2) in the riverhead of the Xilamulun
River. Note that no river water samples are shown in this figure.

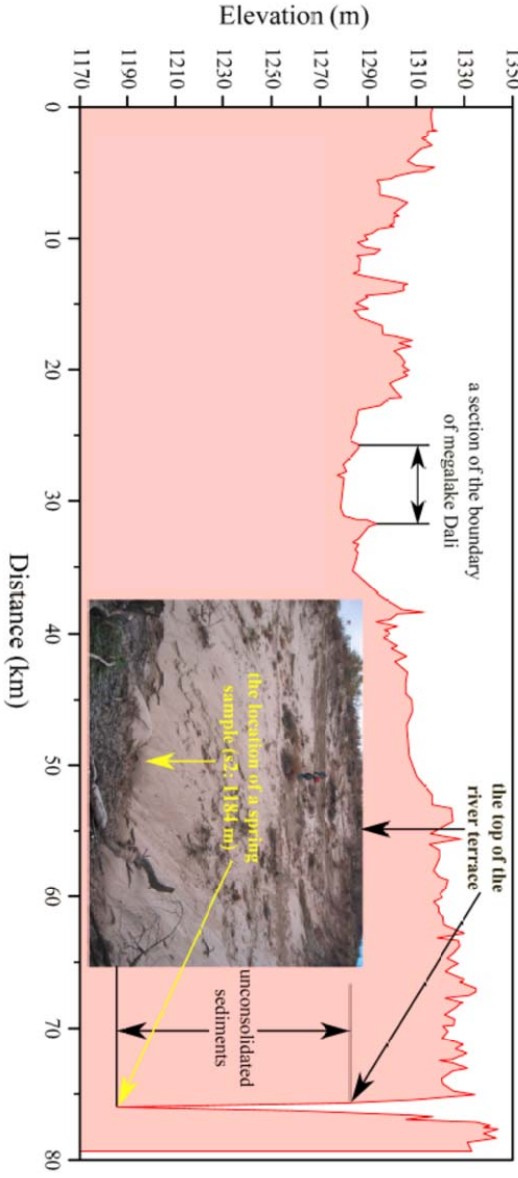

890
891



**Table Captions:**

**Table 1.** The physical parameters measured for the natural water samples in the study area.

| Sample ID | Water type | Latitude (N, degree) | Longitude (E, degree) | Elevation (m a.s.l) | Depth (m) | Temperature (°C) | pH | Eh (mV) | EC (µS/cm) | TDS (mg/L) | Salinity (%) | Alkalinity (meq/L) | Hardness (°dH) |
|---|---|---|---|---|---|---|---|---|---|---|---|---|---|
| g1 | Groundwater | 42.736306 | 116.747333 | 1396 | 12 | 5.8 | 6.72 | 3 | 769 | 410 | 0.6 | 5.47 | 9.42 |
| g2 | Groundwater | 42.736306 | 116.747333 | 1396 | 26 | 6.0 | 6.91 | -10 | 736 | 393 | 0.5 | 4.07 | 12.0 |
| g3 | Groundwater | 42.760194 | 116.760139 | 1355 | 32 | 7.7 | 6.88 | -6 | 725 | 384 | 0.5 | 2.39 | 11.9 |
| g4 | Groundwater | 42.759694 | 116.760417 | 1360 | 7 | 10.0 | 6.74 | 1 | 725 | 387 | 0.5 | 2.20 | 12.3 |
| g5 | Groundwater | 42.759556 | 116.760556 | 1362 | 27 | 7.6 | 6.46 | 16 | 691 | 368 | 0.5 | 2.23 | 15.6 |
| g6 | Groundwater | 42.760111 | 116.760250 | 1365 | 7 | 10.3 | 6.26 | 22 | 1240 | 660 | 0.8 | 3.25 | 24.5 |
| g7 | Groundwater | 42.806361 | 116.747806 | 1352 | 20 | 6.8 | 6.71 | 2 | 297 | 158 | 0.2 | 0.63 | 4.70 |
| g8 | Groundwater | 42.806361 | 116.747806 | 1352 | 16 | 6.5 | 6.92 | -8 | 276 | 147 | 0.2 | 0.58 | 5.00 |
| g9 | Groundwater | 42.850333 | 116.735722 | 1347 | 30 | 7.2 | 6.74 | -1 | 487 | 260 | 0.4 | 3.73 | 12.7 |
| g10 | Groundwater | 42.949861 | 116.759194 | 1321 | 37 | 9.9 | 6.75 | -2 | 337 | 179 | 0.2 | 1.66 | 7.23 |
| g11 | Groundwater | 42.967111 | 116.827528 | 1317 | 60 | 8.6 | 6.99 | -14 | 571 | 302 | 0.4 | 2.40 | 12.9 |
| l1 | Lake water | 42.424611 | 116.769194 | 1368 | / | 16.9 | 9.44 | -151 | 126 | 67 | 0.1 | 0.95 | 1.79 |
| l2 | Lake water | 42.424611 | 116.769194 | 1368 | / | 19.6 | 9.18 | -137 | 132 | 70 | 0.1 | 0.92 | 1.82 |
| l3 | Lake water | 42.424611 | 116.757806 | 1365 | / | 20.2 | 7.38 | -36 | 196 | 105 | 0.1 | 1.53 | 3.36 |
| l4 | Lake water | 42.427083 | 116.757639 | 1366 | / | 20.5 | 7.87 | -64 | 448 | 238 | 0.2 | 3.42 | 6.61 |
| l5 | Lake water | 42.421806 | 116.756917 | 1360 | / | 20.1 | 8.23 | -83 | 173 | 92 | 0.1 | 1.43 | 2.73 |
| l6 | Lake water | 42.736389 | 116.747222 | 1374 | / | 10.7 | 8.35 | -89 | 194 | 103 | 0.1 | 1.53 | 3.30 |
| r1 | River water | 42.530917 | 116.641250 | 1355 | / | 20.6 | 7.31 | -33 | 180 | 96 | 0.1 | 0.88 | 2.23 |
| r2 | River water | 42.310883 | 116.494817 | 1231 | / | 14.9 | 7.67 | -52 | 178 | 95 | 0.1 | 1.21 | 2.50 |
| r3 | River water | 42.385778 | 116.886194 | 1362 | / | 9.5 | 7.62 | -48 | 177 | 94 | 0.1 | 1.45 | 2.62 |
| r4 | River water | 42.931417 | 117.585306 | 1217 | / | 10.5 | 7.97 | -69 | 474 | 252 | 0.3 | 3.22 | 8.73 |
| r5 | River water | 43.079083 | 117.457389 | 1006 | / | 12.9 | 7.87 | -62 | 191 | 101 | 0.1 | 1.37 | 2.88 |
| s1 | Spring water | 42.530917 | 116.641250 | 1359 | / | 20.9 | 6.63 | 5 | 165 | 88 | 0.1 | 0.40 | 1.81 |
| s2 | Spring water | 42.965417 | 116.975361 | 1184 | / | 19.0 | 7.47 | -46 | 371 | 195 | 0.2 | 1.07 | 6.40 |
| p1 | Precipitation | 42.330750 | 116.551694 | 1260 | / | 20.2 | 4.61 | 109 | 78 | 42 | 0.0 | / | 0.61 |

**Table 2.** The concentrations of major cations and anions measured for the water samples in the study area.





| Sample | $F^-$ (mg/L) | $Cl^-$ (mg/L) | $NO_2^-$ (mg/L) | $NO_3^-$ (mg/L) | $SO_4^{2-}$ (mg/L) | $CO_3^{2-}$ (mg/L) | $HCO_3^-$ (mg/L) | $Li^+$ (mg/L) | $Na^+$ (mg/L) | $NH_4^+$ (mg/L) | $K^+$ (mg/L) | $Mg^{2+}$ (mg/L) | $Ca^{2+}$ (mg/L) |
|---|---|---|---|---|---|---|---|---|---|---|---|---|---|
| g1 | 0.13 | 7.90 | 2.32 | 0.48 | 16.1 | 0.00 | 335 | 0.02 | 13.8 | 10.5 | 4.59 | 15.5 | 41.8 |
| g2 | 0.21 | 10.2 | 0.00 | 6.15 | 70.6 | 0.10 | 248 | 0.02 | 13.4 | 6.56 | 3.45 | 17.9 | 56.0 |
| g3 | 0.11 | 79.6 | 0.00 | 0.00 | 141 | 0.00 | 145 | 0.01 | 17.9 | 2.28 | 1.76 | 17.1 | 57.3 |
| g4 | 0.10 | 86.9 | 0.00 | 5.73 | 165 | 0.00 | 134 | 0.02 | 18.0 | 0.00 | 2.02 | 18.5 | 57.3 |
| g5 | 0.07 | 84.8 | 0.00 | 0.76 | 169 | 0.00 | 136 | 0.00 | 39.7 | 1.02 | 2.72 | 20.9 | 76.9 |
| g6 | 0.07 | 141 | 0.00 | 111 | 229 | 0.00 | 198 | 0.00 | 79.8 | 0.00 | 29.47 | 29.3 | 126.7 |
| g7 | 0.37 | 16.3 | 0.00 | 306 | 32.0 | 0.00 | 38.7 | 0.06 | 7.83 | 0.00 | 3.09 | 6.21 | 23.4 |
| g8 | 0.29 | 14.3 | 0.00 | 35.5 | 29.9 | 0.00 | 35.5 | 0.02 | 16.2 | 0.11 | 3.38 | 6.44 | 25.1 |
| g9 | 0.10 | 3.66 | 0.15 | 1.19 | 71.6 | 0.00 | 227 | 0.06 | 12.9 | 0.55 | 4.50 | 14.1 | 67.5 |
| g10 | 0.24 | 18.8 | 0.00 | 49.5 | 9.97 | 0.00 | 101 | 0.00 | 18.5 | 0.00 | 2.09 | 7.92 | 38.7 |
| g11 | 0.28 | 4.94 | 0.00 | 0.00 | 182 | 0.00 | 146 | 0.05 | 20.4 | 2.59 | 2.06 | 13.3 | 70.6 |
| l1 | 0.16 | 3.15 | 0.00 | 0.07 | 4.32 | 0.00 | 57.9 | 0.01 | 5.42 | 0.00 | 0.86 | 3.24 | 7.49 |
| l2 | 0.16 | 3.30 | 0.00 | 1.66 | 4.57 | 0.00 | 55.8 | 0.00 | 5.33 | 0.00 | 0.84 | 3.29 | 7.61 |
| l3 | 0.11 | 3.27 | 0.00 | 0.61 | 2.33 | 0.00 | 93.3 | 0.01 | 5.88 | 0.00 | 1.19 | 5.68 | 14.7 |
| l4 | 0.17 | 22.1 | 0.00 | 0.39 | 3.04 | 0.10 | 208 | 0.00 | 9.21 | 0.70 | 24.2 | 14.1 | 24.2 |
| l5 | 0.09 | 6.24 | 0.00 | 0.65 | 2.97 | 0.10 | 86.8 | 0.01 | 6.72 | 0.00 | 1.16 | 4.91 | 11.4 |
| l6 | 0.18 | 4.29 | 0.00 | 0.80 | 9.34 | 0.10 | 93.0 | 0.01 | 8.41 | 0.00 | 1.36 | 6.47 | 13.0 |
| r1 | 0.30 | 5.76 | 0.00 | 2.38 | 26.7 | 0.30 | 52.4 | 0.01 | 7.15 | 0.00 | 2.99 | 3.41 | 10.3 |
| r2 | 0.19 | 4.82 | 0.00 | 0.65 | 16.4 | 0.10 | 73.1 | 0.01 | 6.82 | 0.00 | 1.92 | 3.96 | 11.4 |
| r3 | 0.64 | 5.46 | 0.00 | 0.43 | 5.57 | 0.00 | 88.1 | 0.01 | 7.11 | 0.00 | 1.13 | 4.04 | 12.1 |
| r4 | 1.08 | 20.4 | 0.00 | 19.3 | 37.3 | 0.50 | 195 | 0.01 | 13.0 | 0.00 | 1.96 | 11.9 | 42.8 |
| r5 | 0.19 | 4.10 | 0.00 | 1.08 | 15.6 | 0.00 | 82.6 | 0.01 | 6.71 | 0.00 | 2.08 | 4.38 | 13.4 |
| s1 | 0.16 | 6.44 | 0.00 | 1.95 | 34.3 | 0.00 | 24.3 | 0.02 | 6.56 | 0.00 | 1.62 | 2.92 | 8.10 |
| s2 | 0.05 | 0.98 | 0.00 | 0.45 | 17.2 | 0.00 | 64.9 | 0.02 | 9.87 | 0.00 | 3.32 | 9.10 | 30.8 |
| p1 | 0.61 | 2.90 | 0.00 | 9.46 | 12.7 | 0.00 | 0.00 | 0.00 | 2.09 | 2.07 | 1.64 | 0.88 | 2.95 |

**Table 3.** The analytical data of stable and radioactive isotopes measured for the water samples in this study.

895
896
897
898





| Sample ID | δD (‰) | σ‰ | δ$^{18}$O (‰) | σ‰ | deuterium excess (d) | Tritium ($^3$H) (TU) |
|---|---|---|---|---|---|---|
| g1 | -66.7 | 0.199 | -8.90 | 0.026 | 4.50 | / |
| g2 | -64.8 | 0.291 | -9.34 | 0.039 | 9.93 | / |
| g3 | -63.4 | 0.269 | -8.64 | 0.008 | 5.66 | / |
| g4 | -66.1 | 0.149 | -9.62 | 0.062 | 10.9 | 7.25 |
| g5 | -65.5 | 0.111 | -9.80 | 0.027 | 13.0 | 9.98 |
| g6 | -68.9 | 0.287 | -10.5 | 0.039 | 15.2 | 22.9 |
| g7 | -73.1 | 0.298 | -10.7 | 0.041 | 12.2 | / |
| g8 | -73.7 | 0.220 | -11.0 | 0.037 | 14.5 | 19.6 |
| g9 | -72.5 | 0.181 | -11.0 | 0.015 | 15.8 | 24.3 |
| g10 | -74.4 | 0.201 | -11.1 | 0.026 | 14.7 | 18.7 |
| g11 | -75.9 | 0.340 | -11.3 | 0.015 | 14.2 | 1.86 |
| l1 | -53.1 | 0.229 | -6.55 | 0.002 | -0.704 | / |
| l2 | -50.7 | 0.304 | -6.32 | 0.026 | -0.161 | / |
| l3 | -42.9 | 0.239 | -4.29 | 0.034 | -8.55 | / |
| l4 | -34.2 | 0.243 | 0.381 | 0.040 | -37.2 | / |
| l5 | -45.1 | 0.206 | -4.99 | 0.009 | -5.16 | / |
| l6 | -52.9 | 0.187 | -6.15 | 0.049 | -3.67 | / |
| r1 | -66.2 | 0.118 | -10.1 | 0.015 | 14.4 | / |
| r2 | -65.0 | 0.148 | -9.55 | 0.012 | 11.4 | / |
| r3 | -73.8 | 0.315 | -11.1 | 0.021 | 14.9 | / |
| r4 | -85.2 | 0.244 | -11.8 | 0.005 | 9.09 | / |
| r5 | -75.0 | 0.195 | -10.1 | 0.003 | 5.69 | / |
| s1 | -70.8 | 0.074 | -10.3 | 0.007 | 11.9 | / |
| s2 | -72.6 | 0.281 | -10.5 | 0.046 | 11.1 | / |
| p1 | -47.4 | 0.374 | -7.14 | 0.017 | 9.69 | / |

**Table 4.** The statistical frequency of rainfall events being >20 mm per year during the recent 30 years from 1985 to 2014. The data come from the China Meteorological Data





Sharing Service System.

| Station | One time/year | Two times/year | Three times/year | Four times/year | Five times/year | Six times/year | Seven times/year | Mean times/year |
|---|---|---|---|---|---|---|---|---|
| Duolun | 2 | 8 | 8 | 4 | 4 | 3 | 1 | 3.4 |
| Xilinhaote | 8 | 5 | 2 | 6 | 3 | 2 | 0 | 2.5 |


**Table 5.** The measured contents of tritium in the groundwater samples studied and the calculated ages of these samples.


| Sample-ID | Tritium content (T.U.) | Possible ages (years) |
|---|---|---|
| g1 | not measured | not clear |
| g2 | not measured | not clear |
| g3 | not measured | not clear |
| g4 | 7.25 | 20-40 |
| g5 | 9.97 | 13-33 |
| g6 | 22.9 | 0-20 |
| g7 | not measured | not clear |
| g8 | 19.6 | 0-20 |
| g9 | 24.3 | 0-17 |
| g10 | 18.7 | 0-22 |
| g11 | 1.86 | 40-65 |
