# Peer review of "Direct or indirect recharge on groundwater in the"

_Hydrology and Earth System Sciences, 2018_

## Referee Comment (RC1) · Anonymous Referee #1 · 20 Nov 2018

The objective of this manuscript is the understanding of groundwater recharge under arid conditions. The authors provide a detailed discussion on the different assumptions they made to explain this recharge. However, this discussion is mainly based on geochemical data including isotopes. To my knowledge, the developed methodology is not new, or, in other words, the authors did not sufficiently highlight the originality of the methodology. Furthermore, the discussion lacks of hydrological considerations. For example, the measured concentrations are the result of the mixing of water moving in the aquifer and the water coming from the recharge. The resulting concentration depends on the different water fluxes which have to be estimated for proper interpretations. Moreover, the travel time in the unsaturated zone has to be discussed in detail. It can be of several decades under these climatic conditions for a groundwater depth

up to 60m. For these reasons, the paper should not be accepted for HESS.

---

## Referee Comment (RC2) · Anonymous Referee #2 · 21 Nov 2018

The work consists in a hydrogeochemical and isotopical study to determine the origin of the groundwater in a sandy desert region of China, bordered by a flat steppe terrain and mountainous regions. The authors propose an accurate analysis of the isotopes and ion chemistries of different water samples including natural samples collected from precipitation, depression spring, shallow and deep aquifers, perpetual lakes and outflowing rivers. They conclude that the groundwater in this desert is possible to originate from remote mountain areas and that the linkage between the desert area and the mountain region is crucial.They use multiple environmental tracers, taking into account the topography, the geomorphology, the climate, the vegetation and the soil, the geology and the hydrology of the Otindag Desert and its surrounding areas. However, in my opinion, the methodology they use for the hydrogeochemical investiga-

tion/interpretation, the key of the study, is not innovative or firstly applied in this kind of problem. Moreover, the outcomes derived (basically) merely by isotopical investigation are not supported by any hydrogeological model. A proper model could take into account and quantify in an integrated scenario the complexity of interacting physical phenomena such as the flow and transport processes, the spreading the mixing and the diffusion of the investigated tracers and the variable (in time and space) recharge in such heterogenous domain. Therefore, I do not think this work is now prompt for publication in HESS.

---

## Author Comment (AC1) · 26 Jan 2019

Authors' response to Hess-2018-395-RC1 (Hess-2018-395, by Bing-Qi Zhu et al., words in blue color)

Author's response: The author's response should be structured in a clear and easy-to-follow sequence: (1) comments from referees/public, (2) author's response, and (3) author's changes in manuscript. Regarding author's changes, a marked-up manuscript version (track changes in Word, latexdiff in LaTeX) converted into a *.pdf including the author's response must be submitted.

(1) Hess-2018-395-RC1 Interactive comment on "Direct or indirect recharge on groundwater in the middle-latitude desert of Otindag, China?" by Bing-Qi Zhu et al. Anony-

mous Referee #1. The objective of this manuscript is the understanding of groundwater recharge under arid conditions. The authors provide a detailed discussion on the different assumptions they made to explain this recharge. However, this discussion is mainly based on geochemical data including isotopes. To my knowledge, the developed methodology is not new, or, in other words, the authors did not sufficiently highlight the originality of the methodology. Furthermore, the discussion lacks of hydrological considerations. For example, the measured concentrations are the result of the mixing of water moving in the aquifer and the water coming from the recharge. The resulting concentration depends on the different water fluxes which have to be estimated for proper interpretations. Moreover, the travel time in the unsaturated zone has to be discussed in detail. It can be of several decades under these climatic conditions for a groundwater depth up to 60m. For these reasons, the paper should not be accepted for HESS.

Authors response and Author's changes in manuscript: We thank the anonymous Referee #1 very much for his/her help in reviewing and commenting on our manuscript. According to above comments, we have revised the original manuscript. First of all, we think that we need to explain to the reviewer #1 here about the question on the research method of this study: (1) Up to now, no any geoscientist has done any research work on groundwater recharge and its sources in the Otindag Desert before, and our research work is the first and pioneering one; (2) because no any study work has been done for groundwater research in this extensive area before, therefore, our study have no any existing information and data of predecessors (especially hydrogeological and hydrological data) to refer to, so we use the traditional methods to carry out preliminary research; However, both the collection of samples, the acquisition of analytical data and the research results based on these methods are the first achievements in this blank area, which are very valuable and also pioneering. Secondly, as to the hydrological problems raised by the reviewer #1, we have actually considered them in the initial sage of this research work. We would like to make the following explanations: (1) About surface water, almost all of these rivers observed in the field are intermittent in space and in time and there is no any hydrological data available for these rivers to be used or referenced by us, so there is no hydrological data for discussion in this study; However, in order to remedy this problem, we are currently conducting field monitoring work in order to obtain these hydrological data; at present, however, we have no systematic data obtained and we can only carry out systematic discussions after obtaining the data in the late years; (2) about groundwater, being similar to surface water, there is no previous hydrological data available for reference in the study area; in order to obtain dynamic hydrological data of groundwater in the study area, at present we are also conducting real-time monitoring of groundwater level in the field. Due to the reason that currently there is no systematic data to discuss, we can only carry out systematic discussions after the data are obtained in the late years. In this study we believe that although the lack of hydrological data is regrettable, it will not conceal the correctness and validity of our discussions using geochemical and isotope geochemical data to explore the recharge sources of groundwater in the Otindag Desert. Thirdly, about the questions of the mixing process, dissolution process, chemical concentrations of groundwater and their water flux, saturation and unsaturation of groundwater proposed by the reviewer, we believe that they are essentially related to the water-rock interaction between groundwater and surrounding rocks, i.e., the speciation modeling and hydrogeological modeling. Therefore, in view of the above problems, we have added new discussions about the processes of water-rock interaction, mechanism of groundwater recharge and related hydrogeological modeling for groundwater in the study area in the discussion part of the revised manuscript. The detailed contents of these revision are also shown here as follows: 4. 6. Speciation modeling and hydrogeological conceptual model Speciation modeling. Selected results of speciation modeling are provided in Table 6. All samples are undersaturated with respect to calcite, aragonite, dolomite, halite and gypsum. The values of log $PCO_2$, ranging between -4.77 and -1.45 in the samples from the sedimentary sandy aquifer, indicating that groundwater in the study area is not at equilibrium with atmospheric $PCO_2$. Based on the above analyses, a conceptual model of groundwater recharge was suggested to facilitate understanding of the hydrogeological conditions in the study area. Local and regional modern precipitation is a negligible source. Quaternary unconsolidated sediments with large exposed area form the main aquifer in the study area. Groundwater is recharged by cold water from remote mountain areas, and it flows from east to west along the Solonker Suture Zone. Evaporation is a minor process during groundwater hydrogeochemical evolution. Mineral dissolution may contribute to groundwater salinization, because saturation indices of all minerals are less than zero, indicating that these minerals still can dissolve into groundwater. These clues mean that the origin of groundwater in the desert is maily controlled by geological structures and processes. The tectonic settings are more important than climatic and topographical settings to explain the origin of groundwater in the desert. In a view of orogenic belt of the global middle-latitude regions, various groundwater and hydrogeological case studies have established a link between geological perspectives and origin of groundwater flows. Tague and Grant (2004) identify, for instance, the dominant control of a young volcanic geological unit on the groundwater regime of the studied region in Oregon, this geological formation having an exceptionally high permeability. Pfister et al. (2017) show that bedrock permeability significantly influences the ratio between average summer and winter run-off of 16 investigated catchments in Luxembourg. For a selection of Swiss catchments, Naef et al. (2015) associate lower groundwater flow with slowly draining porous bedrock and low streamflow during dry periods for catchments dominated by Moraine deposits. Kaser and Hunkeler (2016) have shown that alluvial aquifers, even if they represent only a small portion of the catchment surface, can contribute significantly to the catchment groundwater outflow especially during low-flow periods. Alluvial aquifers can thus also be relevant for total catchment groundwater storage. Chen and Wang (2009) proposed that earthquake is a possible mechanism for groundwater releasing in the Qilian Mountains and discharging it in the Hexi Corridor. Carlier et al. (2018) statistically analyzed 22 catchments of the Swiss Plateau and Prealpes to establish relationships between streamflow indicators and various geological and hydrogeological properties of the bedrock and Quaternary deposits, along with meteorological, soil, land use, and topographical characteristics. The study shows that the geological characteristics dominate catchment response during high and low groundwater flow conditions. These studies focused the influence of base/surrounding rock, topography, recharge source and permeability on groundwater flow in orogeny area. According to the hydrologically active bedrock hypothesis (Uchida et al. 2008) the bedrock is an active reservoir that significantly contributes to baseflow (Tague and Grant 2004; Andermann et al. 2012; Welch and Allen 2012; Birkel et al. 2014). The hydraulic conductivity of the bedrock controls storage processes (Hale et al. 2016; Pfister et al. 2017). Most importantly, the ratio of the hydraulic conductivity to recharge rates has been shown to be relevant for water table elevation (Gleeson and Manning 2008). Haitjema and Mitchell-Bruker (2005) propose a criterion based on the Dupuit-Forchheimer approximation combining this ratio with geometrical aquifer properties and topographical characteristics to determine whether the water table is controlled by the topography or the recharge. From the above review it can be seen that various studies have used spatially distributed, synthetic groundwater models to identify and explore how topography, recharge and/or bedrock permeability influence groundwater fluxes and flow patterns (e.g., Gleeson and Manning 2008; Welch et al. 2012; Welch and Allen 2012; Welch and Allen 2014). These studies highlight the complex interplay of topography and hydrogeology on groundwater flow. They, however, mainly focus on the geology of the bedrock, no studies mentioned the important role of tectonic structure on the groundwater flow. Thus, based on this study in the Otindag Desert, we proposed a simple conceptual model of multiprocesses that constrain the mechanism of groundwater recharge in the desert, namely mountain water (M) – tectonic fault hydrology (T) – unconfined vadose zone with underlying buried fault (V) – groundwater formation and recharge (G), i.e. the MTVG mechanism. Although the model is still conceptual but not practical at present, it provides a new perspective into the origin and evolution of groundwater resources in the middle-latitude deserts of the arid Asia.

Thank the reviewers and the editor of HESS again for your help in dealing with our manuscript. We look forward to hearing from you at your earliest convenience. Best regards, B.Q. Zhu 25 Jan 2019   Comments from previous Referee #1 (hess-2018-71)
Dear Dr/Professor Referee #1: On behalf of my co-authors, we thank you very much for giving us an opportunity to revise our manuscript. We appreciate you very much for your positive and constructive comments and suggestions on our manuscript (hess-2018-71). We have studied your comments carefully and have made revision which marked in red in the revised manuscript. We tried our best to revise our manuscript according to the comments point by point. Attached please find the revised version, which we would like to submit for your kind consideration. Thank you and best regards.

1) The datasets belong to sampling campaigns carried out in different moments (years) and seasons and for this reason in my opinion cannot be discussed together, without a clear distinction between the different phases. Our response: AGREE AND NO CHANGES MADE. Firstly, we thank you very much for this comment from you and we truly agree this point that water samples collected in different moments (years) and seasons cannot be discussed together without a clear distinction between the different water phases. In fact, although we stated in the manuscript that our fieldwork had taken place during the summer season of 2011 and the spring season of 2012, we collected the natural water samples at the same time for the same phases in the study area. For example, (1) all the groundwater samples discussed in this paper were collected during the 2011 summer in five days in the Otindag Desert. For other natural water samples discussed in this study, the detailed sampling methods are as follow: (2) all the spring water samples and (3) the precipitation water sample (p1) discussed in this paper were also collected during the 2011 summer in five days in the study area, and (4) all the river water samples and (5) lake water samples were collected during the spring season of 2012 in three days in the study area. This is to say that the water samples within the same phase are discussed together in the paper.

2) A reconstruction of the piezometric morphology as well as a stratigraphy of the considered study areas should be reported. This could help also the discussion of the groundwater preferential pathways. Our response: AGREE AND CHANGES MADE. We thank you very much for this comment. And yes, according to this comment, we revised the manuscript and focused on reporting the geological (tectonic, lithological, sedimentological and structural), geomorphological, hydrogeological and stratigraphical settings of the study area. Please see the section 2 "Regional setting" of the revised manuscript in its pages 3-5 lines 103-189.

3) The organization of the paper is still at a draft level, since there is not a clear distinction between the results and discussion paragraphs. Many paragraphs need to be summarized and better explained. Our response: AGREE AND CHANGES MADE. We thank you very much for this comment. And yes, we have revised the manuscript accordingly. The structure and content of the paper has been thoroughly reorganized in the revised manuscript, especially for the results and discussion sections, to make the content and context of the paper being more logic, coherent and readable. And yes, almost all of the paragraphs in the paper are newly summarized and explained. The detailed changes can be easily observed in the revised manuscript by reading one of the two resubmitted MS-Word files with the "changes marked" version (in contrast, another version is "clear copy").

4) The number of figures should be reduced (probably putting together some and deleting others). Our response: AGREE AND CHANGES MADE. We thank you very much for this comment. And yes, we have revised the manuscript accordingly. We reduced the number of figures in the revised manuscript by putting some figures together and deleting several figures. At last the revised manuscript has 11 figures compared with the original manuscript that including 15 figures. For example, the Figs. 5, 11, 13, 14a in the original manuscript are deleted in the revised manuscript, and the Figs. 7 and 8, the Figs. 10, 12 and 14a are combined, respectively. In addition, two newly-built figures are added into the revised manuscript according to the second comment from you (the detailed content of this comment can be seen above). The specific changes and the final results of these figures can be seen in the newly submitted revised manuscript.

5) The English is very poor and there are many typo errors. The reported delta notation is wrong. Our response: AGREE AND CHANGES MADE. We thank you very much for this comment. We are very sorry for our poor and incorrect English writing in the original manuscript. For the shortcomings of the English presentation and the grammatical edit in the first paper, we have checked and revised the whole manuscript carefully to avoid language errors, and finally we have got the help of a native English speaking professional to check and improve the English quality of the revised manuscript. We believe that the language is now acceptable for the publishing purpose. In addition, the wrong use of the dalta notation in the original manuscript, such as $\delta2H$, has been corrected as "$\delta D$" in the revised manuscript.

6) Due to the consideration of these main points the manuscript can be accepted only if major revision will be reported. Our response: AGREE AND CHANGES MADE. Special thanks to you for your good comments. We have tried our best to improve the manuscript and made specific changes in the revised manuscript according to the comments from you one by one. These changes will not influence the content and framework of the paper. And here we did not list the changes but marked in red in the revised paper. We hope that the correction will meet with approval. Once again, thank you very much for your comments and suggestions.

  Comments from previous Referee #2 (hess-2018-71)
ward the waters originated from the precipitation on Daxinganling Ranges. Hence, an "indirect" recharge is the main mechanism supporting the water availability in the study arid lands. Two are the main weaknesses of this ms: 1) the chemical/isotopic investigations seem not supported by a (at least minimum) knowledge of the hydrogeological setting. This is likely one of the reasons why the analyses carried out by the authors are mainly able to exclude recharge mechanisms, but not definitely explain from where this water is originated. The last part of Section 5.5 provides a list of speculative mechanisms (lines 614-652): how the Xilamulun river can recharge the Dali lake when Fig. 15 shows that the bed of the former is less elevated than that of the latter? What support the "speculation" about the "flash floods" in the southern portion of the desert? How you only "theoretically estimate" the isotopic firm of the precipitation on the Yinshan Ranges? 2) the contribution is over-long. The introduction addresses the topic with a too-wide perspective, concepts are repeated, with verbose descriptions. There are also too many figures that can be fruitfully combined. The English form must be improved too. Moreover, the location of the study area is unclear: Fig 1a is obscure, the various portions of the desert are not provided in the maps shown in Figs. 1b and 2, a large part of the toponymy cited in the text is not added to the maps. Because of this, the ms need a major revision. Interactive comment on Hydrol. Earth Syst. Sci. Discuss., https://doi.org/10.5194/hess-2018-71, 2018.

The authors' responses to the comments from Referee #2

Dear Dr/Professor Referee #2: On behalf of my co-authors, we thank you very much for giving us an opportunity to revise our manuscript. We appreciate you very much for your positive and constructive comments and suggestions on our manuscript (hess-2018-71). We have read your comments carefully and have made revision which marked in red in the revised manuscript. We tried our best to revise our manuscript according to your comments and suggestions one by one. Attached please find the revised version, which we would like to submit for your kind consideration. Thank you and best regards.

1) The chemical/isotopic investigations seem not supported by a (at least minimum) knowledge of the hydrogeological setting. This is likely one of the reasons why the analyses carried out by the authors are mainly able to exclude recharge mechanisms, but not definitely explain from where this water is originated. The last part of Section 5.5 provides a list of speculative mechanisms (lines 614-652): how the Xilamulun river can recharge the Dali lake when Fig. 15 shows that the bed of the former is less elevated than that of the latter? What support the "speculation" about the "flash floods" in the southern portion of the desert? How you only "theoretically estimate" the isotopic firm of the precipitation on the Yinshan Ranges? Our response: AGREE AND CHANGES MADE. We thank you very much for this comment. Yes, any chemical and isotopic investigations need to be supported by knowledge of the regional- and local-scale hydrogeological settings. According to this comment, we have added the specific information about the hydrogeological, geological (tectonic, lithological, sedimentological and structural), geomorphological, stratigraphical settings of the study area in the revised manuscript. Detailed changes and the added information can be seen from the section "2. Regional settings" and the section "4.5 remote water recharge on groundwater in the Otindag: mountains waters" in the revised manuscript (pages 3-5 lines 103-189 and pages 12-13 lines 442-484). Besides, two newly-built figures about the geological and hydrogeological maps of the study area are also provided as auxiliary instructions to illustrate the hydrogeological characteristics of the Otindag Desert in the revised manuscript. These figures are Figs. 2 and 3 in the revised manuscript. With the help of these newly-added materials we believe that we can definitely and logically explain from where the groundwater in the Otindag is originated. About the Fig. 15 in the original manuscript (at present it is Fig. 11 in the revised manuscript) and the question "how the Xilamulun river can recharge the Dali lake when Fig. 15 shows that the bed of the former is less elevated than that of the latter?", our explanation is that: actually, the elevation of the Xilamulun river channel is not lower than the Dali lake. The recent elevation of the Dali Lake is 1,226 m above sea level (Xiao et al., 2008, J Paleolimnol, 40, 519-528). The elevations of the river samples collected from the
Xilamulun River in this study ranges between 1360 and 1374 m (Table 1). The real elevation data (measured by handheld GPS in the field) for the river samples l1, l2, l3, l4, l5, l6 in this study are 1368 m, 1368m, 1365 m, 1366 m, 1360 m and 1374 m (Table 1), respectively. Thus, the elevation of the Xilamulun river channel is about 140 m higher than that of the Dali Lake. In Fig. 15 (Fig. 11 in the revised manuscript), it shows the variation of the topographical elevation along the section S1 (see Fig. 1b) from the upstream of the Dali Lake to the location site of the spring water samples s2. It does not show the elevations of the river samples from the Xilamulun River. Strictly speaking, however, this sketch map (Fig. 15) is likely to cause misunderstanding if we think about the river water but not the spring water. So we specially stated that "Note that no river water samples are shown in this figure" in the figure caption of Fig. 11 in the revised manuscript. About the question "What support the "speculation" about the "flash floods" in the southern portion of the desert?", we have added specific information about the hydrological settings of the flash floods derived from the Yinshan Piedmont in the section "2. Regional settings" in the revised manuscript (see page 5 lines 158-189). About the question "How you only "theoretically estimate" the isotopic firm of the precipitation on the Yinshan Ranges?", we use the words "theoretically estimate" because we have not obtained the precipitation water samples from the Yinshan Mountains in this study. Thus the isotopic firm of the precipitation on the Yinshan Ranges is calculated based on the altitude effect of mountain temperature on stable isotopes fractionation in the original manuscript. It is thus a theoretical estimation. In order to avoid ambiguity, we deleted the discussion of this "theoretically estimation" in the revised manuscript.

2) The contribution is over-long. The introduction addresses the topic with a too-wide perspective, concepts are repeated, with verbose descriptions. There are also too many figures that can be fruitfully combined. The English form must be improved too. Our response: AGREE AND CHANGES MADE. We thank you very much for this comment. Yes, according to the comment that "the contribution is over-long", we have rewritten the manuscript and made an intensive compression on the length of the paper. At present the number of text words in the revised manuscript has been greatly decreased compared with the original manuscript. According to the comment that "The introduction addresses the topic with a too-wide perspective, concepts are repeated, with verbose descriptions", we have rewritten the introduction section of the manuscript to make the topic being specific and not being too broad in its perspective. We tried our best to avid repeat and verbose descriptions in the revised manuscript whatever on the concept or the context of this section. The detailed changes can be seen in pages 1-3 lines 32-101 in the revised manuscript. According to the comment that "There are also too many figures that can be fruitfully combined", we reduced the number of figures in the revised manuscript by putting some figures together and deleting several figures. At last the revised manuscript has 11 figures compared with the original manuscript that including 15 figures. For example, the Figs. 5, 11, 13, 14a in the original manuscript are deleted in the revised manuscript, and the Figs. 7 and 8, the Figs. 10, 12 and 14a are combined, respectively. In addition, two newly-built figures are added into the revised manuscript according to the first comment from you (the detailed content of this comment can be seen above). The specific changes and the final results of these figures can be seen in the newly submitted revised manuscript. About the comment that "The English form must be improved too", we are very sorry for our poor and incorrect English writing in the original manuscript. For the shortcomings of the English presentation and the grammatical edit in the first paper, we have checked and revised the whole manuscript carefully to avoid language errors, and finally we have got the help of a native English speaking professional to check and improve the English quality of the revised manuscript. We believe that the language is now acceptable for the publishing purpose.

Moreover, the location of the study area is unclear: Fig 1a is obscure, the various portions of the desert are not provided in the maps shown in Figs. 1b and 2, a large part of the toponymy cited in the text is not added to the maps. Our response: AGREE AND CHANGES MADE. We thank you very much for this comment. According to this comment, we have revised the Fig. 1a and 1b and Fig. 2 (now it is Fig. 4 in the revised manuscript) to make them clear and make sure that the various portions of the Otindag Desert are provided in the corresponding maps. We tried our best to add each of the toponymy cited in the text to be included in these maps. The specific changes and the final results of these figures can be seen in the newly submitted revised manuscript (Figs. 1-4).

Finally, we want to say that special thanks to you for your good comments. We have tried our best to improve the manuscript and made specific changes in the revised manuscript according to the comments from you one by one. These changes will not influence the content and framework of the paper. And here we did not list the changes but marked in red in the revised paper. We hope that the correction will meet with approval. Once again, thank you very much for your comments and suggestions.

Please also note the supplement to this comment:
https://www.hydrol-earth-syst-sci-discuss.net/hess-2018-395/hess-2018-395-AC1-supplement.pdf

**Supplement:**

Anonymous Referee #1.

The objective of this manuscript is the understanding of groundwater recharge under arid conditions. The authors provide a detailed discussion on the different assumptions they made to explain this recharge. However, this discussion is mainly based on geochemical data including isotopes. To my knowledge, the developed methodology is not new, or, in other words, the authors did not sufficiently highlight the originality of the methodology. Furthermore, the discussion lacks of hydrological considerations. For example, the measured concentrations are the result of the mixing of water moving in the aquifer and the water coming from the recharge. The resulting concentration depends on the different water fluxes which have to be estimated for proper interpretations. Moreover, the travel time in the unsaturated zone has to be discussed in detail. It can be of several decades under these climatic conditions for a groundwater depth up to 60m. For these reasons, the paper should not be accepted for HESS.

Authors response and Author's changes in manuscript:
We thank the anonymous Referee #1 very much for his/her help in reviewing and commenting on our manuscript. According to above comments, we have revised the original manuscript. First of all, we think that we need to explain to the reviewer #1 here about the question on the research method of this study: (1) Up to now, no any geoscientist has done any research work on groundwater recharge and its sources in the Otindag Desert before, and our research work is the first and pioneering one; (2) because no any study work has been done for groundwater research in this extensive area before, therefore, our study have no any existing information and data of predecessors (especially hydrogeological and hydrological data) to refer to, so we use the traditional methods to carry out preliminary research; However, both the collection of samples, the acquisition of analytical data and the research results based on these methods are the first achievements in this blank area, which are very valuable and also pioneering.
Secondly, as to the hydrological problems raised by the reviewer #1, we have actually considered them in the initial sage of this research work. We would like to make the following explanations: (1) About surface water, almost all of these rivers observed in the field are intermittent in space and in time and there is no any hydrological data available for these rivers to be used or referenced by us, so there is no hydrological data for discussion in this study; However, in order to remedy this problem, we are currently conducting field monitoring work in order to obtain these hydrological data; at present, however, we have no systematic data obtained and we can only carry out systematic discussions after obtaining the data in the late years; (2) about groundwater, being similar to surface water, there is no previous hydrological data available for reference in the study area; in order to obtain dynamic hydrological data of groundwater in the study area, at present we are also conducting real-time monitoring of groundwater level in the field. Due to the reason that currently there is no systematic data to discuss, we can only carry out systematic discussions after the data are obtained in the late years. In this study we believe that although the lack of hydrological data is regrettable, it will not conceal the correctness and validity of our discussions using geochemical and isotope geochemical data to explore the recharge sources of groundwater in the Otindag Desert.
Thirdly, about the questions of the mixing process, dissolution process, chemical concentrations of groundwater and their water flux, saturation and unsaturation of groundwater proposed by the reviewer, we believe that they are essentially related to the water-rock interaction between groundwater and surrounding rocks, i.e., the speciation modeling and hydrogeological modeling. Therefore, in view of the above problems, we have added new discussions about the processes of water-rock interaction, mechanism of groundwater recharge and related hydrogeological modeling for groundwater in the study area in the discussion part of the revised manuscript. The detailed contents of these revision are also shown here as follows:

4. 6. Speciation modeling and hydrogeological conceptual model

Speciation modeling. Selected results of speciation modeling are provided in Table 6. All samples are undersaturated with respect to calcite, aragonite, dolomite, halite and gypsum. The values of log PCO2, ranging between -4.77 and -1.45 in the samples from the sedimentary sandy aquifer, indicating that groundwater in the study area is not at equilibrium with atmospheric PCO2.

Based on the above analyses, a conceptual model of groundwater recharge was suggested to facilitate understanding of the hydrogeological conditions in the study area. Local and regional modern precipitation is a negligible source. Quaternary unconsolidated sediments with large exposed area form the main aquifer in the study area. Groundwater is recharged by cold water from remote mountain areas, and it flows from east to west along the Solonker Suture Zone. Evaporation is a minor process during groundwater hydrogeochemical evolution. Mineral dissolution may contribute to groundwater salinization, because saturation indices of all minerals are less than zero, indicating that these minerals still can dissolve into groundwater. These clues mean that the origin of groundwater in the desert is maily controlled by geological structures and processes. The tectonic settings are more important than climatic and topographical settings to explain the origin of groundwater in the desert.

In a view of orogenic belt of the global middle-latitude regions, various groundwater and hydrogeological case studies have established a link between geological perspectives and origin of groundwater flows. Tague and Grant (2004) identify, for instance, the dominant control of a young volcanic geological unit on the groundwater regime of the studied region in Oregon, this geological formation having an exceptionally high permeability. Pfister et al. (2017) show that bedrock permeability significantly influences the ratio between average summer and winter run-off of 16 investigated catchments in Luxembourg. For a selection of Swiss catchments, Naef et al. (2015) associate lower groundwater flow with slowly draining porous bedrock and low streamflow during dry periods for catchments dominated by Moraine deposits. Kaser and Hunkeler (2016) have shown that alluvial aquifers, even if they represent only a small portion of the catchment surface, can contribute significantly to the catchment groundwater outflow especially during low-flow periods. Alluvial aquifers can thus also be relevant for total catchment groundwater storage. Chen and Wang (2009) proposed that earthquake is a possible mechanism for groundwater releasing in the Qilian Mountains and discharging it in the Hexi Corridor. Carlier et al. (2018) statistically analyzed 22 catchments of the Swiss Plateau and Prealpes to establish relationships between streamflow indicators and various geological and hydrogeological properties of the bedrock and Quaternary deposits, along with meteorological, soil, land use, and topographical characteristics. The study shows that the geological characteristics dominate catchment response during high and low groundwater flow conditions.

These studies focused the influence of base/surrounding rock, topography, recharge source and permeability on groundwater flow in orogeny area. According to the hydrologically active bedrock hypothesis (Uchida et al. 2008) the bedrock is an active reservoir that significantly contributes to baseflow (Tague and Grant 2004; Andermann et al. 2012; Welch and Allen 2012; Birkel et al. 2014). The hydraulic conductivity of the bedrock controls storage processes (Hale et al. 2016; Pfister et al. 2017). Most importantly, the ratio of the hydraulic conductivity to recharge rates has been shown to be relevant for water table elevation (Gleeson and Manning 2008). Haitjema and Mitchell-Bruker (2005) propose a criterion based on the Dupuit-Forchheimer approximation combining this ratio with geometrical aquifer properties and topographical characteristics to determine whether the water table is controlled by the topography or the recharge. From the above review it can be seen that various studies have used spatially distributed, synthetic groundwater models to identify and explore how topography, recharge and/or bedrock permeability influence groundwater fluxes and flow patterns (e.g., Gleeson and Manning 2008; Welch et al. 2012; Welch and Allen 2012; Welch and Allen 2014).

These studies highlight the complex interplay of topography and hydrogeology on groundwater flow. They, however, mainly focus on the geology of the bedrock, no studies mentioned the important role of tectonic structure on the groundwater flow. Thus, based on this study in the Otindag Desert, we proposed a simple conceptual model of multiprocesses that constrain the mechanism of groundwater recharge in the desert, namely mountain water (M) – tectonic fault hydrology (T) – unconfined vadose zone with underlying buried fault (V) – groundwater formation and recharge (G), i.e. the MTVG mechanism. Although the model is still conceptual but not practical at present, it provides a new perspective into the origin and evolution of groundwater resources in the middle-latitude deserts of the arid Asia.

Thank the reviewers and the editor of HESS again for your help in dealing with our manuscript.
We look forward to hearing from you at your earliest convenience.
Best regards,
B.Q. Zhu
Jan 2019
The manuscript describes interesting results about the recharge mechanisms of arid zones in China, especially considering the importance of the topic. Despite the multidisciplinary approach, which is very useful in groundwater recharge studies, there are many weak points which have to be improved for a publication in HESS. The main points are listed below: 1) The datasets belong to sampling campaigns carried out in different moments (years) and seasons and for this reason in my opinion cannot be discussed together, without a clear distinction between the different phases. 2) A reconstruction of the piezometric morphology as well as a stratigraphy of the considered study areas should be reported. This could help also the discussion of the groundwater preferential pathways. 3) The organization of the paper is still at a draft level, since there is not a clear distinction between the results and discussion paragraphs. Many paragraphs need to be summarized and better explained. 4) The number of figures should be reduced (probably putting together some and deleting others). 5) The English is very poor and there are many typo errors. The reported delta notation is wrong. Due to the consideration of these main points the manuscript can be accepted only if major revision will be reported.

**The authors' responses to the comments from Referee #1**

Dear Dr/Professor Referee #1:

On behalf of my co-authors, we thank you very much for giving us an opportunity to revise our manuscript. We appreciate you very much for your positive and constructive comments and suggestions on our manuscript (hess-2018-71). We have studied your comments carefully and have made revision which marked in red in the revised manuscript. We tried our best to revise our manuscript according to the comments point by point. Attached please find the revised version, which we would like to submit for your kind consideration. Thank you and best regards.

1) The datasets belong to sampling campaigns carried out in different moments (years) and seasons and for this reason in my opinion cannot be discussed together, without a clear distinction between the different phases.

Our response: AGREE AND NO CHANGES MADE.

Firstly, we thank you very much for this comment from you and we truly agree this point that water samples collected in different moments (years) and seasons cannot be discussed together without a clear distinction between the different water phases. In fact, although we stated in the manuscript that our fieldwork had taken place during the summer season of 2011 and the spring season of 2012, we collected the natural water samples at the same time for the same phases in the study area. For example, (1) all the groundwater samples discussed in this paper were collected during the 2011 summer in five days in the Otindag Desert. For other natural water samples discussed in this study, the detailed sampling methods are as follow: (2) all the spring water samples and (3) the precipitation water sample (p1) discussed in this paper were also collected during the 2011 summer in five days in the study area, and (4) all the river water samples and (5) lake water samples were collected during the spring season of 2012 in three days in the study area. This is to say that the water samples within the same phase are discussed together in the paper.

2) A reconstruction of the piezometric morphology as well as a stratigraphy of the considered study areas should be reported. This could help also the discussion of the groundwater preferential pathways.

Our response: AGREE AND CHANGES MADE.

We thank you very much for this comment. And yes, according to this comment, we revised the manuscript and focused on reporting the geological (tectonic, lithological, sedimentological and structural), geomorphological, hydrogeological and stratigraphical settings of the study area. Please see the section 2 "Regional setting" of the revised manuscript in its pages 3-5 lines 103-189.

3) The organization of the paper is still at a draft level, since there is not a clear distinction between the results and discussion paragraphs. Many paragraphs need to be summarized and better explained.

Our response: AGREE AND CHANGES MADE.

We thank you very much for this comment. And yes, we have revised the manuscript accordingly. The structure and content of the paper has been thoroughly reorganized in the revised manuscript, especially for the results and discussion sections, to make the content and context of the paper being more logic, coherent and readable. And yes, almost all of the paragraphs in the paper are newly summarized and explained. The detailed changes can be easily observed in the revised manuscript by reading one of the two resubmitted MS-Word files with the "changes marked" version (in contrast, another version is "clear copy").

4) The number of figures should be reduced (probably putting together some and deleting others).

Our response: AGREE AND CHANGES MADE.

We thank you very much for this comment. And yes, we have revised the manuscript accordingly. We reduced the number of figures in the revised manuscript by putting some figures together and deleting several figures. At last the revised manuscript has 11 figures compared with the original manuscript that including 15 figures. For example, the Figs. 5, 11, 13, 14a in the original manuscript are deleted in the revised manuscript, and the Figs. 7 and 8, the Figs. 10, 12 and 14a are combined, respectively. In addition, two newly-built figures are added into the revised manuscript according to the second comment from you (the detailed content of this comment can be seen above). The specific changes and the final results of these figures can be seen in the newly submitted revised manuscript.

5) The English is very poor and there are many typo errors. The reported delta notation is wrong.

Our response: AGREE AND CHANGES MADE.

We thank you very much for this comment. We are very sorry for our poor and incorrect English writing in the original manuscript. For the shortcomings of the English presentation and the grammatical edit in the first paper, we have checked and revised the whole manuscript carefully to avoid language errors, and finally we have got the help of a native English speaking professional to check and improve the English quality of the revised manuscript. We believe that the language is now acceptable for the publishing purpose.

In addition, the wrong use of the delta notation in the original manuscript, such as $\delta^2H$, has been corrected as "$\delta D$" in the revised manuscript.

6) Due to the consideration of these main points the manuscript can be accepted only if major revision will be reported.

Our response: AGREE AND CHANGES MADE.

Special thanks to you for your good comments. We have tried our best to improve the manuscript and made specific changes in the revised manuscript according to the comments from you one by one. These changes will not influence the content and framework of the paper. And here we did not list the changes but marked in red in the revised paper. We hope that the correction will meet with approval. Once again, thank you very much for your comments and suggestions.
Groundwater availability in arid and semi-arid regions is one of the key issues in hydrogeology and is becoming even more important because of the expected climate changes. Within this context, the contribution by Zhu and Ren provides an interesting analysis on the possible recharge supporting the availability of significant groundwater resources in the Otindag desert, north-eastern China. The analyses have been carried out using hydrogeochemical tracers and isotopic measurements on water samples collected from groundwater, surficial (river, lake, and spring) waters, and precipitation water, as well as in-situ records of temperature, pH, conductivity, and TDS concentration. The various steps implemented by the authors to reject possible hypotheses on the groundwater origin (e.g., water flowing from another nearby arid area, precipitation, paleo-water resources) are presented in detail and discussed. Zhu and Ren concludes that, based on the available evidences, the groundwater resources in this region are recharged by the leakage through the bed on incise rivers bounding the desert to the east and conveying downward the waters originated from the precipitation on Daxinganling Ranges. Hence, an "indirect" recharge is the main mechanism supporting the water availability in the study arid lands.

Two are the main weaknesses of this ms: 1) the chemical/isotopic investigations seem not supported by a (at least minimum) knowledge of the hydrogeological setting. This is likely one of the reasons why the analyses carried out by the authors are mainly able to exclude recharge mechanisms, but not definitely explain from where this water is originated. The last part of Section 5.5 provides a list of speculative mechanisms (lines 614-652): how the Xilamulun river can recharge the Dali lake when Fig. 15 shows that the bed of the former is less elevated than that of the latter? What support the "speculation" about the "flash floods" in the southern portion of the desert? How you only "theoretically estimate" the isotopic firm of the precipitation on the Yinshan Ranges? 2) the contribution is over-long. The introduction addresses the topic with a too-wide perspective, concepts are repeated, with verbose descriptions. There are also too many figures that can be fruitfully combined. The English form must be improved too. Moreover, the location of the study area is unclear: Fig 1a is obscure, the various portions of the desert are not provided in the maps shown in Figs. 1b and 2, a large part of the toponymy cited in the text is not added to the maps. Because of this, the ms need a major revision.

**The authors' responses to the comments from Referee #2**

Dear Dr/Professor Referee #2:
On behalf of my co-authors, we thank you very much for giving us an opportunity to revise our manuscript. We appreciate you very much for your positive and constructive comments and suggestions on our manuscript (hess-2018-71). We have read your comments carefully and have made revision which marked in red in the revised manuscript. We tried our best to revise our manuscript according to your comments and suggestions one by one. Attached please find the revised version, which we would like to submit for your kind consideration. Thank you and best regards.

1) The chemical/isotopic investigations seem not supported by a (at least minimum) knowledge of the hydrogeological setting. This is likely one of the reasons why the analyses carried out by the authors are mainly able to exclude recharge mechanisms, but not definitely explain from where this water is originated. The last part of Section 5.5 provides a list of speculative mechanisms (lines 614-652): how the Xilamulun river can recharge the Dali lake when Fig. 15

shows that the bed of the former is less elevated than that of the latter? What support the "speculation" about the "flash floods" in the southern portion of the desert? How you only "theoretically estimate" the isotopic firm of the precipitation on the Yinshan Ranges?

Our response: AGREE AND CHANGES MADE.

We thank you very much for this comment. Yes, any chemical and isotopic investigations need to be supported by knowledge of the regional- and local-scale hydrogeological settings. According to this comment, we have added the specific information about the hydrogeological, geological (tectonic, lithological, sedimentological and structural), geomorphological, stratigraphical settings of the study area in the revised manuscript. Detailed changes and the added information can be seen from the section "2. Regional settings" and the section "4.5 remote water recharge on groundwater in the Otindag: mountains waters" in the revised manuscript (pages 3-5 lines 103-189 and pages 12-13 lines 442-484). Besides, two newly-built figures about the geological and hydrogeological maps of the study area are also provided as auxiliary instructions to illustrate the hydrogeological characteristics of the Otindag Desert in the revised manuscript. These figures are Figs. 2 and 3 in the revised manuscript. With the help of these newly-added materials we believe that we can definitely and logically explain from where the groundwater in the Otindag is originated.

About the Fig. 15 in the original manuscript (at present it is Fig. 11 in the revised manuscript) and the question "how the Xilamulun river can recharge the Dali lake when Fig. 15 shows that the bed of the former is less elevated than that of the latter?", our explanation is that: actually, the elevation of the Xilamulun river channel is not lower than the Dali lake. The recent elevation of the Dali Lake is 1,226 m above sea level (Xiao et al., 2008, J Paleolimnol, 40, 519-528). The elevations of the river samples collected from the Xilamulun River in this study ranges between 1360 and 1374 m (Table 1). The real elevation data (measured by handheld GPS in the field) for the river samples l1, l2, l3, l4, l5, l6 in this study are 1368 m, 1368m, 1365 m, 1366 m, 1360 m and 1374 m (Table 1), respectively. Thus, the elevation of the Xilamulun river channel is about 140 m higher than that of the Dali Lake. In Fig. 15 (Fig. 11 in the revised manuscript), it shows the variation of the topographical elevation along the section S1 (see Fig. 1b) from the upstream of the Dali Lake to the location site of the spring water samples s2. It does not show the elevations of the river samples from the Xilamulun River. Strictly speaking, however, this sketch map (Fig. 15) is likely to cause misunderstanding if we think about the river water but not the spring water. So we specially stated that "Note that no river water samples are shown in this figure" in the figure caption of Fig. 11 in the revised manuscript.

About the question "What support the "speculation" about the "flash floods" in the southern portion of the desert?", we have added specific information about the hydrological settings of the flash floods derived from the Yinshan Piedmont in the section "2. Regional settings" in the revised manuscript (see page 5 lines 158-189).

About the question "How you only "theoretically estimate" the isotopic firm of the precipitation on the Yinshan Ranges?", we use the words "theoretically estimate" because we have not obtained the precipitation water samples from the Yinshan Mountains in this study. Thus the isotopic firm of the precipitation on the Yinshan Ranges is calculated based on the altitude effect of mountain temperature on stable isotopes fractionation in the original manuscript. It is thus a theoretical estimation. In order to avoid ambiguity, we deleted the discussion of this "theoretical estimation" in the revised manuscript.

2) The contribution is over-long. The introduction addresses the topic with a too-wide perspective, concepts are repeated, with verbose descriptions. There are also too many figures that can be fruitfully combined. The English form must be improved too.

Our response: AGREE AND CHANGES MADE.

We thank you very much for this comment. Yes, according to the comment that "the contribution is over-long", we have rewritten the manuscript and made an intensive compression on the length of the paper. At present the number of text words in the revised manuscript has been greatly decreased compared with the original manuscript.

According to the comment that "The introduction addresses the topic with a too-wide perspective, concepts are repeated, with verbose descriptions", we have rewritten the introduction section of the manuscript to make the topic being specific and not being too broad in its perspective. We tried our best to avid repeat and verbose descriptions in the revised manuscript whatever on the concept or the context of this section. The detailed changes can be seen in pages 1-3 lines 32-101 in the revised manuscript.

According to the comment that "There are also too many figures that can be fruitfully combined", we reduced the number of figures in the revised manuscript by putting some figures together and deleting several figures. At last the revised manuscript has 11 figures compared with the original manuscript that including 15 figures. For example, the Figs. 5, 11, 13, 14a in the original manuscript are deleted in the revised manuscript, and the Figs. 7 and 8, the Figs. 10, 12 and 14a are combined, respectively. In addition, two newly-built figures are added into the revised manuscript according to the first comment from you (the detailed content of this comment can be seen above). The specific changes and the final results of these figures can be seen in the newly submitted revised manuscript.

About the comment that "The English form must be improved too", we are very sorry for our poor and incorrect English writing in the original manuscript. For the shortcomings of the English presentation and the grammatical edit in the first paper, we have checked and revised the whole manuscript carefully to avoid language errors, and finally we have got the help of a native English speaking professional to check and improve the English quality of the revised manuscript. We believe that the language is now acceptable for the publishing purpose.

Moreover, the location of the study area is unclear: Fig 1a is obscure, the various portions of the desert are not provided in the maps shown in Figs. 1b and 2, a large part of the toponymy cited in the text is not added to the maps.

Our response: AGREE AND CHANGES MADE.

We thank you very much for this comment. According to this comment, we have revised the Fig. 1a and 1b and Fig. 2 (now it is Fig. 4 in the revised manuscript) to make them clear and make sure that the various portions of the Otindag Desert are provided in the corresponding maps. We tried our best to add each of the toponymy cited in the text to be included in these maps. The specific changes and the final results of these figures can be seen in the newly submitted revised manuscript (Figs. 1-4).

Finally, we want to say that special thanks to you for your good comments. We have tried our best to improve the manuscript and made specific changes in the revised manuscript according to the comments from you one by one. These changes will not influence the content and framework of the paper. And here we did not list the changes but marked in red in the revised paper. We hope that the correction will meet with approval. Once again, thank you very much for your comments and suggestions.

[revised manuscript text omitted]

**4. 6. Speciation modeling and hydrogeological conceptual model**

Speciation modeling. Selected results of speciation modeling are provided in Table 6. All samples are undersaturated with respect to calcite, aragonite, dolomite, halite and gypsum. The values of log $P_{CO_2}$, ranging between -4.77 and -1.45 in the samples from the sedimentary sandy aquifer, indicating that groundwater in the study area is not at equilibrium with atmospheric $P_{CO_2}$.

Based on the above analyses, a conceptual model of groundwater recharge was suggested to facilitate understanding of the hydrogeological conditions in the study area. Local and regional modern precipitation is a negligible source. Quaternary unconsolidated sediments with large exposed area form the main aquifer in the study area. Groundwater is recharged by cold water from remote mountain areas, and it flows from east to west along the Solonker Suture Zone. Evaporation is a minor process during groundwater hydrogeochemical evolution. Mineral dissolution may contribute to groundwater salinization, because saturation indices of all minerals are less than zero, indicating that these minerals still can dissolve into groundwater. These clues mean that the origin of groundwater in the desert is maily controlled by geological structures and processes. The tectonic settings are more important than climatic and topographical settings to explain the origin of groundwater in the desert.

In a view of orogenic belt of the global middle-latitude regions, various groundwater and hydrogeological case studies have established a link between geological perspectives and origin of groundwater flows. Tague and Grant (2004) identify, for instance, the dominant control of a young volcanic geological unit on the groundwater regime of the studied region in Oregon, this geological formation having an exceptionally high permeability. Pfister et al. (2017) show that bedrock permeability significantly influences the ratio between average summer and winter run-off of 16 investigated catchments in Luxembourg. For a selection of Swiss catchments, Naef et al. (2015) associate lower groundwater flow with slowly draining porous bedrock and low streamflow during dry periods for catchments dominated by Moraine deposits. Kaser and Hunkeler (2016) have shown that alluvial aquifers, even if they represent only a small portion of the catchment surface, can contribute significantly to the catchment groundwater outflow especially during low-flow periods. Alluvial aquifers can thus also be relevant for total catchment groundwater storage. Chen and Wang (2009) proposed that earthquake is a possible mechanism for groundwater releasing in the Qilian Mountains and discharging it in the Hexi Corridor. Carlier et al. (2018) statistically analyzed 22 catchments of the Swiss Plateau and Prealpes to establish relationships between streamflow indicators and various geological and hydrogeological properties of the bedrock and Quaternary deposits, along with meteorological, soil, land use, and topographical characteristics. The study shows that the geological characteristics dominate catchment response during high and low groundwater flow conditions.

These studies focused the influence of base/surrounding rock, topography, recharge source and permeability on groundwater flow in orogeny area. According to the hydrologically active bedrock hypothesis (Uchida et al. 2008) the bedrock is an active reservoir that significantly contributes to baseflow (Tague and Grant 2004; Andermann et al. 2012; Welch and Allen 2012; Birkel et al. 2014). The hydraulic conductivity of the bedrock controls storage processes (Hale et al. 2016; Pfister et al. 2017). Most importantly, the ratio of the hydraulic conductivity to recharge rates has been shown to be relevant for water table elevation (Gleeson and Manning 2008). Haitjema and Mitchell-Bruker (2005) propose a criterion based on the Dupuit-Forchheimer approximation combining this ratio with geometrical aquifer properties and topographical characteristics to determine whether the water table is controlled by the topography or the recharge. From the above review it can be seen that various studies have used spatially distributed, synthetic groundwater models to identify and explore how topography, recharge and/or bedrock permeability influence groundwater fluxes and flow patterns (e.g., Gleeson and Manning 2008; Welch et al. 2012; Welch and Allen 2012; Welch and Allen 2014).

These studies highlight the complex interplay of topography and hydrogeology on groundwater flow. They, however, mainly focus on the geology of the bedrock, no studies mentioned the important role of tectonic structure on the groundwater flow. Thus, based on this study in the Otindag Desert, we proposed a simple conceptual model of multiprocesses that constrain the mechanism of groundwater recharge in the desert, namely mountain water (M) – tectonic fault hydrology (T) – unconfined vadose zone with underlying buried fault (V) – groundwater formation and recharge (G), i.e. the MTVG mechanism. Although the model is still conceptual but not practical at present, it provides a new perspective into the origin and evolution of groundwater resources in the middle-latitude deserts of the arid Asia.

**5. Conclusions**

In the middle-latitude desert zone of northern China, many deserts such as the Otindag and Badanjilin Deserts, are unexpectedly rich in groundwater resources, although they have no surface runoff and have been under an arid or hyper-arid climate for a long period of time. How groundwaters originated and recharged in these deserts are thus key questions that are still under debate. For some earth scientists, the direct recharge is thought to be very important for groundwaters in the wide desert lands of northern China, due to the lack of surface runoffs. However, groundwater availability is very much a function of the local- and regional-scale geological and climatic settings. To achieve an integrated understanding of the groundwater recharge and its controlling mechanisms is of great significance. In this study, groundwater recharge was explored using multiple environmental tracers in the Otindag Desert of northern China, a region that is under the influence of the East Asian Summer Monsoon (EASM) climate. Compared to modern summer precipitation, the groundwaters, river waters and spring waters are depleted in δD and $\delta^{18}$O. All these waters shared a same Craig line, indicating a genetic relationship on their recharge sources. The stable isotopic signals of the groundwaters is more depleted than those of the modern summer precipitation and this suggests that the groundwaters studied could only be sourced from cold water different from the EASM precipitation. In general, the analyses revealed that the highland remote water resources from the Daxing'Anling and Yinshan Mountains were isotopically and geochemically traced to be a major source for the groundwater in the Otindag. It suggests that the modern indirect recharge mechanism, instead of the direct recharge and the palaeo-water recharge, is the most significant for groundwater recharge in the eastern Otindag. This study provides a new perspective into the origin and evolution of groundwater resources in the middle-latitude desert zone of northern China.

**Acknowledgements**

This study was financially supported by the National Natural Science Foundation of China (41771014 ),  the National Key Research and Development Program of China (2016YFA0601900), and the National Natural Science Foundation of China (41602196). We thank the China Meteorological Data Sharing Service system for providing the weather data. Sincere thanks are also extended to Profs. Xiaoping Yang, Xunming Wang, Jule Xiao and other workmates, e.g., Ziting Liu, Hongwei Li, and DeguoZhangfor their generous help in the research work.

[revised manuscript text omitted]

[revised manuscript text omitted]

**4. 6. Speciation modeling and hydrogeological conceptual model**

Speciation modeling. Selected results of speciation modeling are provided in Table 6. All samples are undersaturated with respect to calcite, aragonite, dolomite, halite and gypsum. The values of log $P_{CO2}$, ranging between -4.77 and -1.45 in the samples from the sedimentary sandy aquifer, indicating that groundwater in the study area is not at equilibrium with atmospheric $P_{CO2}$.

Based on the above analyses, a conceptual model of groundwater recharge was suggested to facilitate understanding of the hydrogeological conditions in the study area. Local and regional modern precipitation is a negligible source. Quaternary unconsolidated sediments with large exposed area form the main aquifer in the study area. Groundwater is recharged by cold water from remote mountain areas, and it flows from east to west along the Solonker Suture Zone. Evaporation is a minor process during groundwater hydrogeochemical evolution. Mineral dissolution may contribute to groundwater salinization, because saturation indices of all minerals are less than zero, indicating that these minerals still can dissolve into groundwater. These clues mean that the origin of groundwater in the desert is maily controlled by geological structures and processes. The tectonic settings are more important than climatic and topographical settings to explain the origin of groundwater in the desert.

In a view of orogenic belt of the global middle-latitude regions, various groundwater and hydrogeological case studies have established a link between geological perspectives and origin of groundwater flows. Tague and Grant (2004) identify, for instance, the dominant control of a young volcanic geological unit on the groundwater regime of the studied region in Oregon, this geological formation having an exceptionally high permeability. Pfister et al. (2017) show that bedrock permeability significantly influences the ratio between average summer and winter run-off of 16 investigated catchments in Luxembourg. For a selection of Swiss catchments, Naef et al. (2015) associate lower groundwater flow with slowly draining porous bedrock and low streamflow during dry periods for catchments dominated by Moraine deposits. Kaser and Hunkeler (2016) have shown that alluvial aquifers, even if they represent only a small portion of the catchment surface, can contribute
significantly to the catchment groundwater outflow especially during low-flow
periods. Alluvial aquifers can thus also be relevant for total catchment groundwater
storage. Chen and Wang (2009) proposed that earthquake is a possible mechanism
for groundwater releasing in the Qilian Mountains and discharging it in the Hexi
Corridor. Carlier et al. (2018) statistically analyzed 22 catchments of the Swiss Plateau
and Prealpes to establish relationships between streamflow indicators and various
geological and hydrogeological properties of the bedrock and Quaternary deposits,
along with meteorological, soil, land use, and topographical characteristics. The study
shows that the geological characteristics dominate catchment response during high
and low groundwater flow conditions.

These studies focused the influence of base/surrounding rock, topography,
recharge source and permeability on groundwater flow in orogeny area. According to
the hydrologically active bedrock hypothesis (Uchida et al. 2008) the bedrock is an
active reservoir that significantly contributes to baseflow (Tague and Grant 2004;
Andermann et al. 2012; Welch and Allen 2012; Birkel et al. 2014). The hydraulic
conductivity of the bedrock controls storage processes (Hale et al. 2016; Pfister et al.
2017). Most importantly, the ratio of the hydraulic conductivity to recharge rates has
been shown to be relevant for water table elevation (Gleeson and Manning 2008).
Haitjema and Mitchell-Bruker (2005) propose a criterion based on the
Dupuit-Forchheimer approximation combining this ratio with geometrical aquifer
properties and topographical characteristics to determine whether the water table is
controlled by the topography or the recharge. From the above review it can be seen
that various studies have used spatially distributed, synthetic groundwater models to
identify and explore how topography, recharge and/or bedrock permeability
influence groundwater fluxes and flow patterns (e.g., Gleeson and Manning 2008;
Welch et al. 2012; Welch and Allen 2012; Welch and Allen 2014).

These studies highlight the complex interplay of topography and hydrogeology
on groundwater flow. They, however, mainly focus on the geology of the bedrock, no
studies mentioned the important role of tectonic structure on the groundwater flow.
Thus, based on this study in the Otindag Desert, we proposed a simple conceptual
model of multiprocesses that constrain the mechanism of groundwater recharge in
the desert, namely mountain water (M) – tectonic fault hydrology (T) – unconfined
vadose zone with underlying buried fault (V) – groundwater formation and recharge
(G), i.e. the MTVG mechanism. Although the model is still conceptual but not
practical at present, it provides a new perspective into the origin and evolution of
groundwater resources in the middle-latitude deserts of the arid Asia.

**5. Conclusions**

In the middle-latitude desert zone of northern China, many deserts such as the
Otindag and Badanjilin Deserts, are unexpectedly rich in groundwater resources,
although they have no surface runoff and have been under an arid or hyper-arid
climate for a long period of time. How groundwaters originated and recharged in
these deserts are thus key questions that are still under debate. For some earth
scientists, the direct recharge is thought to be very important for groundwaters in
the wide desert lands of northern China, due to the lack of surface runoffs. However,
groundwater availability is very much a function of the local- and regional-scale geological and climatic settings. To achieve an integrated understanding of the groundwater recharge and its controlling mechanisms is of great significance. In this study, groundwater recharge was explored using multiple environmental tracers in the Otindag Desert of northern China, a region that is under the influence of the East Asian Summer Monsoon (EASM) climate. Compared to modern summer precipitation, the groundwaters, river waters and spring waters are depleted in δD and δ$^{18}$O. All these waters shared a same Craig line, indicating a genetic relationship on their recharge sources. The stable isotopic signals of the groundwaters is more depleted than those of the modern summer precipitation and this suggests that the groundwaters studied could only be sourced from cold water different from the EASM precipitation. In general, the analyses revealed that the highland remote water resources from the Daxing'Anling and Yinshan Mountains were isotopically and geochemically traced to be a major source for the groundwater in the Otindag. It suggests that the modern indirect recharge mechanism, instead of the direct recharge and the palaeo-water recharge, is the most significant for groundwater recharge in the eastern Otindag. This study provides a new perspective into the origin and evolution of groundwater resources in the middle-latitude desert zone of northern China.

**Acknowledgements**

This study was financially supported by the National Natural Science Foundation of China (41771014), the National Key Research and Development Program of China (2016YFA0601900), and the National Natural Science Foundation of China (41602196). We thank the China Meteorological Data Sharing Service system for providing the weather data. Sincere thanks are also extended to Profs. Xiaoping Yang, Xunming Wang, Jule Xiao and other workmates, e.g., Ziting Liu, Hongwei Li, and DeguoZhangfor their generous help in the research work.

[revised manuscript text omitted]